# Engineering an autonomous VH domain to modulate intracellular pathways and to interrogate the eIF4F complex

Yuri Frosi[1,2,12], Yen-Chu Lin[1,3,4,12], Jiang Shimin[1,2,12], Siti Radhiah Ramlan[1,2], Kelly Hew[5,6], Alf Henrik Engman[5,6], Anil Pillai[5,6], Kit Yeung[5,6], Yue Xiang Cheng[5,6], Tobias Cornvik[6], Par Nordlund [5,6,7], Megan Goh [1], Dilraj Lama[8], Zachary P. Gates [2,9], Chandra S. Verma[6,10,11], Dawn Thean[1], David P. Lane[1], Ignacio Asial[5,6] ✉ & Christopher J. Brown[1,2] ✉

An attractive approach to target intracellular macromolecular interfaces and to model putative drug interactions is to design small high-affinity proteins. Variable domains of the immunoglobulin heavy chain (VH domains) are ideal miniproteins, but their development has been restricted by poor intracellular stability and expression. Here we show that an autonomous and disufhide-free VH domain is suitable for intracellular studies and use it to construct a high-diversity phage display library. Using this library and affinity maturation techniques we identify VH domains with picomolar affinity against eIF4E, a protein commonly hyper-activated in cancer. We demonstrate that these molecules interact with eIF4E at the eIF4G binding site via a distinct structural pose. Intracellular overexpression of these miniproteins reduce cellular proliferation and expression of malignancy-related proteins in cancer cell lines. The linkage of high-diversity in vitro libraries with an intracellularly expressible miniprotein scaffold will facilitate the discovery of VH domains suitable for intracellular applications.

Many intracellular macromolecular interfaces exist in human cancers that are highly desirable anti-cancer therapeutic targets, such as Myc:Max, KRAS:RAF and eIF4E:4G[1]. Some of these interactions present significant hurdles to small molecule development due to the nature of their molecular surfaces. eIF4E is a particularly attractive oncogenic target as it forms part of the eIF4F complex, which performs a critical role in mediating cap-dependent protein synthesis[2] and whose components (eIF4E, eIF4A and eIF4G) are frequently mis-regulated in many tumours and often associated with poor prognosis in patients and chemoresistance[3–5]. eIF4E interacts with the m7G cap structure at the 5′ end of mRNAs and recruits the eIF4F complex via eIF4G, which in turn initiates translation. This process regulates the translation of a specific subset of sensitive mRNAs (e.g., c-Myc, CCND1), which are significantly involved in cellular transformation and oncogenesis[6–8].

[1]p53 Laboratory (A*STAR), 8A Biomedical Grove, #06-04/05, Neuros/Immunos 138648, Singapore. [2]Disease Intervention Technology Laboratory (DITL), Institute of Molecular and Cell Biology, A*STAR, Singapore 138673, Singapore. [3]Insilico Medicine Taiwan Ltd., Taipei City 110208, Taiwan. [4]Department of Pharmacy, National Yang Ming Chiao Tung University, Taipei City 112304, Taiwan. [5]DotBio Pte. Ltd., 1 Research Link, Singapore 117604, Singapore. [6]School of Biological Sciences, Nanyang Technological University, 60 Nanyang Drive, 637551 Singapore, Singapore. [7]Department of Oncology and Pathology, Karolinska Institutet, Stockholm 17177, Sweden. [8]Department of Microbiology, Tumor and Cell Biology, Karolinska Institutet, Biomedicum Quarter 7B-C Solnavägen 9, 17165 Solna, Sweden. [9]Institute of Sustainability for Chemicals, Energy and Environment (ISCE2), A*STAR, 8 A Biomedical Grove, #07-01 Neuros Building, 138665 Singapore, Singapore. [10]Bioinformatics Institute (A*STAR), 30 Biopolis Street, #07-01 Matrix, 138671 Singapore, Singapore. [11]Department of Biological Sciences, National University of Singapore, 14 Science Drive 4, 117543 Singapore, Singapore. [12]These authors contributed equally: Yuri Frosi, Yen-Chu Lin, Jiang Shimin. ✉e-mail: ignacio.asial@dotbiopharma.com; cjbrown@imcb.a-star.edu.sg

One approach to circumvent the issues associated with small molecule discovery is to design small high-affinity proteins (approx. 60 to 100 amino acids in size) to target these interfaces, termed miniproteins[9,10]. Miniproteins are of special interest as they can easily be expressed within mammalian cells and engineered into transgenic mice, where they can be induced in a temporal and tissue-specific manner. In the absence of pharmacological-specific reagents they offer a precise tool for the interrogation of fundamental biology and target validation[11]. Miniproteins unlike traditional knock-down and knock out strategies to validate candidate proteins for therapeutic development, more closely model the fundamental interactions of drugs with their targets[12].

Variable domains of the human immunoglobulin heavy chain (VH domains) are ideal candidates for use as miniproteins. They possess three binding loops of variable length (CDR-H1, CDR-H2 and CDR-H3) that are naturally randomised to generate a wide repertoire of binders for antigen recognition by the immune system. The development of these scaffolds as single domain binding reagents for intracellular studies has been restricted by poor stability, due to the loss of stabilizing interactions with the variable light-chain domain in the intact antibody[13,14]. However, single chain variable fragments (ScFv) termed "Intrabodies" consisting of a VH and a VL domain connected by a flexible peptide have been used to probe intracellular targets. Unfortunately, ScFvs are deleteriously affected by the reducing conditions of the cell, which prevent the formation of the VH and VL intra-domain disulfide bonds and consequently hinders their proper folding leading to ScFvs that are non-functional, poorly expressed with short half-lives and poor solubilities[15–18].

Several approaches have been devised to overcome these liabilities and to increase the discovery rate of intrabodies, such as screening intrabody-antigen interactions within cells and the use of predetermined frameworks and consensus sequences with improved solubility and expression properties in vivo[19]. Single domain antibody fragments generated from VH and VL domains have also been reported to express within cells and to retain their functionality[20]. Consensus sequences derived from these intrabodies were unaffected by substitution of the residues responsible for the VH domain intra-domain disulfide bond. The dispensability of the intra-domain disulfide bond is also supported by crystallographic data showing that its absence does not perturb VH and VL domain structures[21]. An alternative antibody-type modality capable of intracellular expression are Nanobodies (VHH)[22,23]. These are variable domains derived from the heavy-chain only *camelid* antibodies that are monomeric and soluble that contain 1 to 2 intra-domain disulfide bonds. However, most Nanobodies yielded from conventional screening approaches are non-functional within living cells[22,24]. To overcome these issues, different groups have reported approaches to improve selection of intracellularly functional binders from synthetic or immunized libraries[22,25,26] and efforts to identify cysteine-free, non-immunoglobulin based miniproteins that are suitable for intracellular expression have also been reported[9].

Motivated by these observations and by data demonstrating that engineered human VH domain scaffolds can exist monomerically in solution[13,27], we endeavour in this work to address the liabilities associated with intracellular expression of ScFvs and other miniproteins by engineering and optimising an autonomous and disulfide-free human VH domain for intracellular expression studies through the use of CoFi[28] and Hot-CoFi[29] directed evolution techniques. We then exploit the optimised VH framework sequence to generate a phage display library with randomised CDR1, 2 and 3 loops and use it to identify several VH domain binders against eIF4E. The isolated VH domains are shown to inhibit the eIF4E:4G interaction and are then further evolved by affinity maturation to generate a picomolar binder against eIF4E (termed VH-S4) suitable for activity modulation studies in mammalian cells (Fig. S1). Failure to regulate the eIF4F complex frequently occurs in cancers when the 4E-binding proteins (4E-BP1,2 and 3) are hyperphosphorylated by mTOR[30–33], and fail to displace eIF4G from eIF4E. Both proteins possess a common primary binding motif (YXXXXLΦ, X = any amino acid and Φ = hydrophobic amino acid) to eIF4E[34]. The high-affinity VH domain interactor shares an overlapping interaction site with peptides derived from this binding motif but interacts though an alternative binding mode. The interface between eIF4G and eIF4E is a potential site of therapeutic development and has been targeted with several small molecule and peptidic strategies[6,35]. However, many of these molecules are poorly active and have not been clinically approved and the alternative interaction pose formed by VH-S4 with eIF4E represents a distinct and unexplored avenue for therapeutic development strategies.

The autonomous VH domain technology presented here offers a rapid and efficient pipeline to discover new binding poses for therapeutic lead development, as well as the identification of modalities that can be used to therapeutically model and validate these sites for drug development in biological systems. This methodology also enables the advantageous linkage of high diversity in vitro libraries ($10^8$–$10^{10}$) with an intracellularly expressible miniprotein scaffold, in contrast to in vivo selection techniques that use smaller libraries ($10^5$–$10^6$), to increase the discovery rate of VH domains suitable for intracellular applications.

## Results

### Generation of a stable, disulfide-free, autonomous antibody variable domain for intracellular expression

The 4D5 VH domain was chosen as the template for stabilization, as its parental IgG trastuzumab is an approved FDA drug with a favourable immunogenic and stability profile[13]. The 4D5 VH domain was fused to an N-terminal signal sequence for periplasmic expression in *E. coli* and a random mutagenesis library of 4D5 created (Table S1). This was used to initiate 2 rounds of selection using Hot-CoFi[29] described in Fig. 1a, resulting in the isolation of 53 stabilized VH domain variants (Fig. 1b). Hot-CoFi is an attractive system for the selection of thermally stable VH variants as high-yield bacterial expression has been shown to correlate well with ScFvs expression in the cytoplasm of mammalian cells[19]. Among the identified variants, VH-33 and VH-36 were the most thermally stable, with $T_{CAGG}$s (midpoint of thermal cellular aggregation curves, see materials and methods) of 72.9 and 73.4 °C, respectively, an increase of over 20 °C in comparison to the 4D5 clone (Fig. 1c and Table S1). The VH-36 variant contained the following mutations, H33D, S93G and W103R. Both H33D and W103R are located at the former VH and VL domain interface and mainly improve the thermal stability of the VH variant by decreasing its hydrophobicity (Fig. S2).

VH-36 was selected as the starting template for evolution of a disulfide-free VH domain. Three residues in VH-36, C22 and C92, as well as the core residue A24, were then randomised to generate a library suitable for screening. A24 was included to facilitate optimal packing of residues in the hydrophobic core of the protein due to its proximity to C22 and C92. The signalling peptide, stII, was also removed from the VH-36 expression construct (termed VH-36i) to ensure cytoplasmic expression and allow screening in a cellular reducing environment. Screening was performed using Hot-CoFi and five unique variants of VH-36i were identified. VH36i.1 possessed the mutations C22S, A24C and C92T (Fig. 1c, d and Table S1). Hydrogen bonding between C22S and C92T most likely occurs to replace the lost disulfide bond.

### Optimization of a disulfide-free autonomous antibody variable domain for phage display

The VH-36i.1 clone and its parent templates (VH-36 and VH-36i) all contained the mutation A100bP (Ala to Pro mutation at position 100b, according to Kabat numbering within the CDR-H3 loop[36]. An undesirable feature that restricts the conformational and diversity space open to randomized CDR-H3 loop regions and in turn decreases their probability of successful target interaction. Therefore, the CDR-H3 region (W95 to P100b) was removed and replaced by the sequence SSSA, creating the new construct VH-37i (Table S1), which had significantly

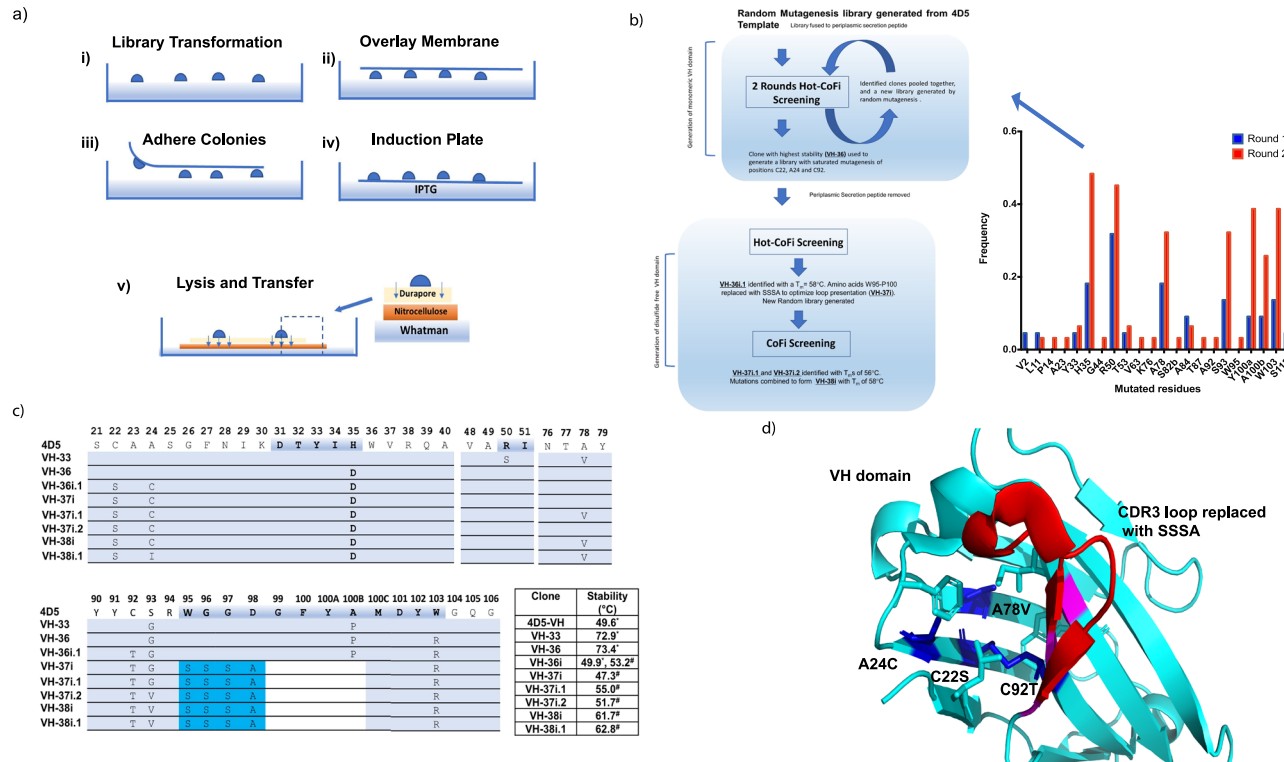

**Fig. 1 | Engineering and optimising an autonomous and disulfide-free human VH domain for phage display and intracellular mammalian expression.**
**a** Schematic of the CoFi selection procedure. (i) Cells transformed with the randomly mutagenized library are plated and grown on LB plates. (ii) The filter membrane is overlaid on top of the colonies, (iii) which adhere to the membrane and can be removed from the LB agar. (iv) Protein expression is induced in the transformed bacterial colonies by placing the filter on IPTG containing LB plates. (v) The cells are lysed by and soluble proteins diffuse through the filter membrane and are captured on the nitrocellulose membrane while inclusion bodies are retained on the filter membrane. Cell lysis is performed by placing the filter and the nitrocellulose membranes in native lysis buffer to release the VH domain clones from the bacterial colonies. By probing the nitrocellulose membrane with antibodies, colonies expressing soluble proteins can be identified. If performing the Hot CoFi method, the induction plate is incubated at the desired temperature above the wild-type protein melting temperature at step iv), before transferring the filter membrane onto the top of the nitrocellulose membrane in step v). **b** Flow-chart describing the directed evolution procedure to generate a disulfide-free

autonomous VH Domain. Inset: Frequencies of specific mutations isolated from the first 2 rounds of Hot-Cofi selection. Source data is provided in the Source Data File. **c** VH clones isolated in the directed evolution procedure outlined in **b** with their respective mutations and associated thermal stabilities. The VH domain clones are numbered using the Kabat nomenclature system to denote residues and CDR positions (highlighted in blue in the 4D5 sequence). Only sections of the aligned VH domains containing mutations are shown. Residues highlighted in cyan were inserted to replace residues 95–100B in clones VH-37i and onwards. White space denotes residues deleted from the VH domain sequence as result of SSSA insertion. **d** Structure of 4D5 VH domain, extracted from the crystal structure of Trastuzumab (PDB code: 6BAE), highlighting the location of the stabilising core mutations found in VH-38i (blue) and CDR3 loop replacement (red). Magenta shows location of mutations on former VH:VL interface (Fig. S2). Figure 1a is reprinted by permission from Springer Nature: Nature Methods, Colony filtration blot: a new screening method for soluble protein expression in *Escherichia coli*, Cornvik T., Dahlroth S.L., Magnusdottir A., Herman M.D., Knaust R., Ekberg M. and Nordlund P., Copyright © 2005.

reduced thermal stability ($T_m$ = 47.3 °C). VH-37i, was then used to create a random mutagenesis library for CoFi screening[28] to compensate for these liabilities. CoFi screening as opposed to Hot-CoFi was performed at room temperatures to identify soluble and colloidally stable variants[28]. From this process four variants were identified with VH-37i.1 and VH-37i.2 exhibiting the highest thermal stabilities ($T_m$), approximately 55.0 and 51.7 °C respectively. Both clones contained a single mutation with VH-37i.1 bearing a A78V mutation in the core of the VH domain, whilst VH-37i.2 contained the mutation G93V located near the CDR-H3 loop (Fig. 1d and Table S1). The introduction of A78V most likely leads to improved packing interactions in the hydrophobic core of the protein around C22S and C92T, whilst the S93V substitution likely increases the rigidity of the β-strand leading into the CDR3 region and thus stabilising the VH framework also. These mutations were combined to generate VH38i with an improved $T_m$ of approximately 61.7 °C. To remove the final cysteine at position 24, it was either mutated to the Ala residue found in 4D5, Ile, Val, Tyr, or Trp. The VH-38i variant VH38i.1, containing mutation C24I was the most stable with improved $T_m$ of 62.8 °C (Fig. 1c). Replacement of C24 with these residues lead to further optimised hydrophobic packing of residues around C22S and C92T

(Fig. 1d). The amino-acid sequence of each VH variant is available in Table S1, and their corresponding stability assessment curves in Fig. S3.

## VH domain phage display library construction and delineation of the VH-1A2 and VH-1C5 interaction sites on eIF4E

A library of VH domains were then displayed on the pIII protein of M13 phage, where the CDR1 and CDR2 of the VH-38i.1 template were randomized conservatively, whilst CDR3 was randomized with sequences of different lengths ranging from 10 to 24 residues and biased towards residues serine and tyrosine with ~20–25% and 10–15% frequency, respectively (Fig. S4), resulting in a library with 2.87 × 10^10 unique clones (see methods and materials). The valine introduced at position 93 in VH-37i.2 and retained in VH-38i.1 was substituted back to a glycine to improve CDR3 loop dynamics and improve potential epitope presentation. Three rounds of phage display selection against purified human full-length eIF4E, followed by screening of 95 individually picked phage clones by ELISA led to the identification of nine unique VH domains (Table S2). VH-1A2 and VH-1C5 were validated as the highest affinity binders with $K_d$s of 115.2 ± 4.4 nM and 154.3 ± 70.3 nM, respectively (Fig. 2a and Table 1). The selected CDR loop sequences of the

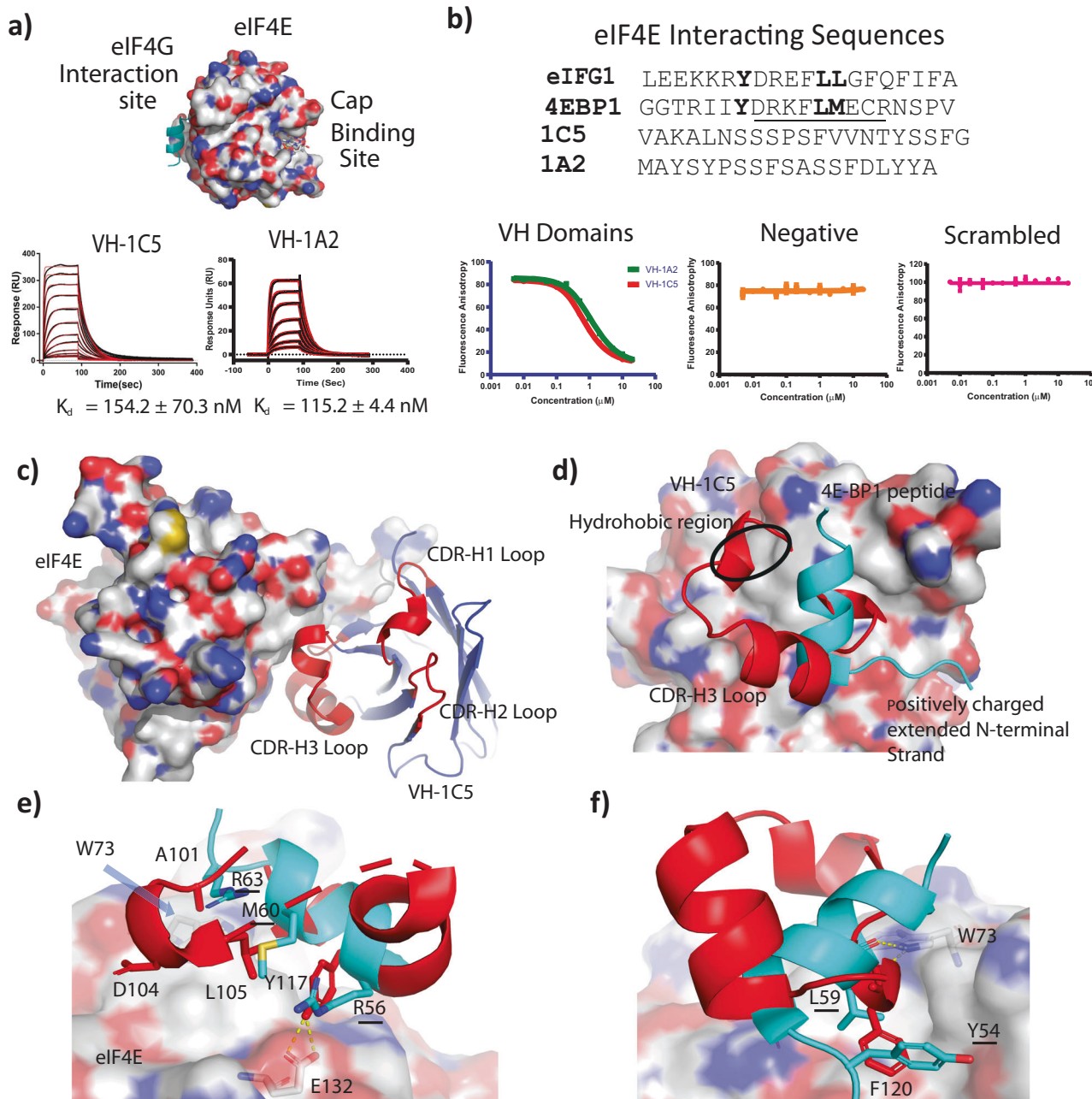

**Fig. 2 | Structural characterization studies delineating eIF4E VH domain binding sites. a** Locations of the cap-binding and eIF4G interaction sites on eIF4E. Surface plasmon resonance sensograms showing 2-fold titration series of VH-1C5 and VH-1A2 from 2.5 μM to 25 nM against eIF4E immobilised via amine coupling on CM5 chips, respectively. Derived $K_d$s are shown on the graphs (mean ± standard error) and are calculated from $n = 2$ independent experiments). **b** Comparison of naturally occurring eIF4E interacting sequences (4E-BP1 and eIF4G1) with the selected VH domain library sequences (1A2 and 1C5). Graphs show representative titrations of VH domains 1C5 and 1A2 against eiF4E in fluorescent polarization competition experiments using a FAM labelled eIF4G1 peptide (Ac-KKRYSRDFL-LALQK-(FAM)). Derived IC50s are shown in the graph (mean ± standard error) and were calculated from $n = 2$ independent experiments. Source data is provided in the Source Data File. **c** Crystal structure of eIF4E complexed with VH-1C5. The phage library selected CDR3 loop insert that interacts with eIF4E is highlighted in red, whilst the main fold of the VH domain is shown in cyan. Highlighted in orange are close contacts between the VH domain and eIF4E that lie outside the library insert region of the CDR3 loop. **d** Overlay of the CDR3 region (red) with the eIF4E interacting sequence from 4E-BP1 (cyan). Both sequences adopt very different secondary structural folds to each other and have overlapping interaction sites with eIF4E. The VH-1C5 CDR3 loop forms additional contacts with a hydrophobic region on the surface of eIF4E that the 4E-BP1 peptide does not, whilst the positively charged N-terminal extended strand of 4E-BP1 instead Interacts with a negatively charged patch on eIF4E. **e** Residues A101 and D104 of VH-1C5 form contacts with the hydrophobic region of eIF4E, whilst L105 recapitulates the interactions of the M60 residue from 4E-BP1. Y117 from the CDR3 loop of VH-1C5 forms a hydrogen bond with E13, replacing the electrostatic interaction observed with R56 in 4E-BP1. **f** The F120 residue found in VH-1C5 mimics the hydrophobic contacts made by L59 and Y54 of 4E-BP1 against eIF4E. In addition, the conserved hydrogen bond formed between the carbonyl backbone of L59 with the indole side chain of W73 is repeated by the carbonyl backbone of F220.

**Table 1 | Binding and kinetics parameters of various VH domains against eIF4E**

| | $K_d$ (nM) | $k_{on}$ ($S^{-1}M^{-1}$) | $k_{off}$ ($S^{-1}$) |
|---|---|---|---|
| VH-1A2 | 115.2 ± 4.4 | $3.43 \times 10^5 \pm 3.75 \times 10^4$ | $3.78 \times 10^{-2} \pm 3.17 \times 10^{-3}$ |
| VH-1C5 | 154.3 ± 70.3 | $2.85 \times 10^5 \pm 8.5 \times 10^4$ | $4.11 \times 10^{-2} \pm 6.99 \times 10^{-3}$ |
| VH-1C5$^{D104A}$ (VH-M4) | 4.8 ± 1.7 | $2.46 \times 10^6 \pm 3.89 \times 10^4$ | $1.18 \times 10^{-2} \pm 3.90 \times 10^{-3}$ |
| VH-1C5$^{D104A/S108R}$ | 0.94 ± 0.024 | $2.63 \times 10^6 \pm 2.98 \times 10^5$ | $2.48 \times 10^{-3} \pm 3.79 \times 10^{-4}$ |
| VH-1C5$^{D104A/F120I}$ (VH-S2) | 0.64 ± 0.06 | $1.18 \times 10^6 \pm 1.85 \times 10^5$ | $7.64 \times 10^{-4} \pm 1.80 \times 10^{-4}$ |
| VH1-C5$^{D104A/S108R/F120I}$ (VH-S4) | 0.057 ± 0.004 | $5.18 \times 10^6 \pm 8.55 \times 10^5$ | $2.96 \times 10^{-4} \pm 6.47 \times 10^{-5}$ |
| 4E-BP1$^{4ALA}$ (1:1) | 0.60 ± 0.14 | $1.05 \times 10^{10} \pm 1.48 \times 10^9$ | 5.93 ± 0.56 |
| | | $k_{on1}, k_{on2}$ | $k_{off1}, k_{off2}$ |
| 4E-BP1$^{4ALA}$ (2 state) | 0.68 ± 0.13 | $8.35 \times 10^6 \pm 4.90 \times 10^6$, $1.70 \times 10^{-2} \pm 1.08 \times 10^{-2}$ | $3.19 \times 10^{-4} \pm 7.91 \times 10^{-5}$, $1.84 \times 10^{-4} \pm 7.00 \times 10^{-5}$ |

$K_d$, $k_{on}$ and $k_{off}$ values were derived from SPR experiments using either single cycle or multi cycle injection experiments against eIF4E amine coupled to a CM5 sensorchip. Parameters were derived using a 1:1 site binding model. However, with regards to 4E-BP1$^{4ALA}$, binding parameters were initially determined using a 1:1 site binding model, which generated a poor fit and physically irrelevant $K_{on}$ and $K_{off}$ values beyond the detection limit of the machine. This analysis was superseded with a 2-state analysis, which generated more meaningful kinetic parameters. The use of a 2-state model is supported by the following reasons: (1) 4E-BP1$^{4ALA}$ undergoes a disorder to order transition upon binding eIF4E and, (2) it interacts at eIF4E through 2 binding sites. Binding and kinetic data values (mean ± standard error) shown in the table were calculated from $n = 2$ independent experiments. Data shown for 4E-BP1$^{4ALA}$ was calculated from $n = 3$ independent experiments. Source data is provided in the Source Data File.

VH-1A2 and VH-1C5 domains shared little similarity to the shared interacting motif of eIF4E binding proteins such as eIF4G1 and the 4E-BP family (YXXXXLΦ) (Fig. 2b)[34]. Competitive-based fluorescence anisotropy experiments were performed that mapped the binding of VH-1A2 and VH-1C5 to the eIF4G interaction site (Fig. 2b).

The crystal complex of eIF4E and VH-1C5 (Fig. S5) revealed that VH-1C5 bound eIF4E at a position that overlaps with the eIF4G interaction site. The CDR3 loop (residues 11-120) of VH-1C5 forms a folded structure that interacts with eIF4E, whose structure contrasts with the 'L'-shaped conformers formed by 4E-BP1 and eIF4G derived linear peptides when in complex with eIF4E (Fig. 2c, d)[34]. The random coil section (residues 101–107) of the CDR3 loop orientates the VH domain residue L105 into a position that mimics the interactions made by the conserved hydrophobic residue in the eIF4E interaction (residues M60 and L630 in 4E-BP1 and eIF4G1, respectively) (Fig. 2e). The random coil section (residues 101–104) of the VH domain also forms additional contacts with a hydrophobic region located on eIF4E unexploited by the eIFG1 and 4E-BP1 derived peptides (Fig. 2e)[37,38] Y117, located on the helical segment (residues 108–117) of the CDR3 loop, replaces the electrostatic interaction mediated by R55 and R625 of the 4E-BP1 and eIF4G1 peptides with E132 by forming hydrophobic packing interactions with L135 and a hydrogen bond with E132 on the surface of eIF4E (Fig. 2e). F220, positioned on the short helical turn motif (residues 119–121) that precedes the loop re-joining the VH domain, forms several hydrophobic contacts with the eIF4E residues L39, V69 and I138 and a backbone hydrogen bond interaction with the indole side chain of W73 on eIF4E (Fig. 2f), which replicated the interactions formed by the conserved Y and L residues of the eIF4E interaction motif (Y54/624 and L59/629 of 4E-BP1 and eIF4G1, respectively).

**Alanine scanning mutagenesis reveals VH-1C5 residues distal to the interaction interface are critical for binding**
Alanine mutagenesis scanning of the CDR3 loop revealed two classes of mutants that adversely affected the binding of VH-1C5 to eIF4E (Fig. 3a, b). The first class of mutants describe residues identified that are in direct contact with eIF4E (K102, A103, D104, L105, T116, Y117 and F120). Replacement of D104 with alanine (Fig. 3a) induced an -17-fold improvement in affinity (Fig. 3c and Table 1) with a $K_d$ of 4.8 ± 1.7 nM in the VH-1C5 variant (VH-M4). The second class of mutations describe the alanine substitution of residues V101, P110, V113, V114 and F122 that abolished eIF4E binding but were not located at the VH-1C5 interface. These residues form a distinct hydrophobic cluster that interact with several hydrophobic residues located on the β-sheet face of the VH domain (V39, L47 and W49) on the former VL interface (Fig. 3d). This is highly similar to CDR-H3 loop interactions that occur in a variety of Nanobodies, where an extended CDR3 loop folds back on to the protein

to interact with and shield hydrophobic residues from the solvent (Fig. S6)[22,39,40]. The sensitivity of VH-1C5 binding to eIF4E when this cluster is mutated confirms that these residues ensure the correct folding of the CDR3 loop. A salt bridge further rigidifies the CDR3 loop that forms between residues R52 and D37 at the CDR3 loop and the VH framework interface, which orientates R52 to form electrostatic interactions with the CDR3 backbone carbonyls of S107 and S108 (Fig. 3e).

**Evolution of an ultra high-affinity miniprotein inhibitor of the eIF4E:4G interaction**
The VH-M4 binder ($K_d$ = 4.8 ± 1.7 nM) was used as a template sequence to generate a randomly mutagenized library for affinity maturation to identify improved binders against eIF4E using yeast display (Fig. 4a). Three rounds of kinetic selection were performed with increasing incubations times of 8, 60 and 110 min to increase the competitive pressure of unlabelled eIF4E against the VH domain library complexed to fluorescent eIF4E (Fig. 4b). The final round of kinetic selection resulted in the identification of only three eIF4E binders: VH-1C5$^{D104A/S108R/F120I}$, VH-1C5$^{D104A/S24G/F120I}$ and VH-1C5$^{D104A/Y97C/F120I}$. The VH-1C5$^{D104A/S108R/F120I}$ clone (termed VH-S4) dominated the final round with a frequency > 90%, which bound eIF4E with a mid-picomolar $K_d$ of 0.057 ± 0.004 nM (Fig. 4c and Table S3). The two substitutions identified in the VH-S4 sequence were then individually introduced into the VH-1C5-M4 clone to generate VH-1C5$^{D104A/S108R}$ and VH-1C5$^{D104A/F120I}$ (termed VH-S2). These bound eIF4E with $K_d$s of 0.94 ± 0.04 nM and 0.64 ± 0.06 nM, respectively (Fig. 4c and Table 1).

The crystal structure of eIF4E: VH-1C5$^{D104A/S108R/F120I}$ (VH-S4) complex revealed that the F120I mutation improved the hydrophobic contacts between the two proteins (Fig. 4d and Fig. S7). The structure also showed that the S108R mutation does not interact directly with the surface of eIF4E, but that it engages a structured water network that forms an h-bond to the backbone carbonyl of R128 on eIF4E (Fig. 4e and Fig. S5). In addition, S108R formed a cation-π interaction with the sidechain of the neighbouring F112, further stabilising the local fold of the CDR3 loop. The eIF4E: VH-1C5$^{D104A/S108R/F120I}$ (VH-S4) structure also revealed that the substitution D104A, removes the unfavourable packing of the negatively charged D104 side chain against Y76 (Fig. 4f and Fig. S7).

**VH-S4 disrupts eIF4F complex formation and cap-dependent translation in vitro**
VH-S4 immuno-precipitated higher amounts of endogenous eIF4E than the more weakly binding VH domain variants (VH-1C5, VH-M4 and VH-S2) (Fig. 5a). A VH-1C5 negative control (VH-1C5$^{SCRM}$), where the CDR3 loop residues were scrambled, also failed to pull-down eIF4E

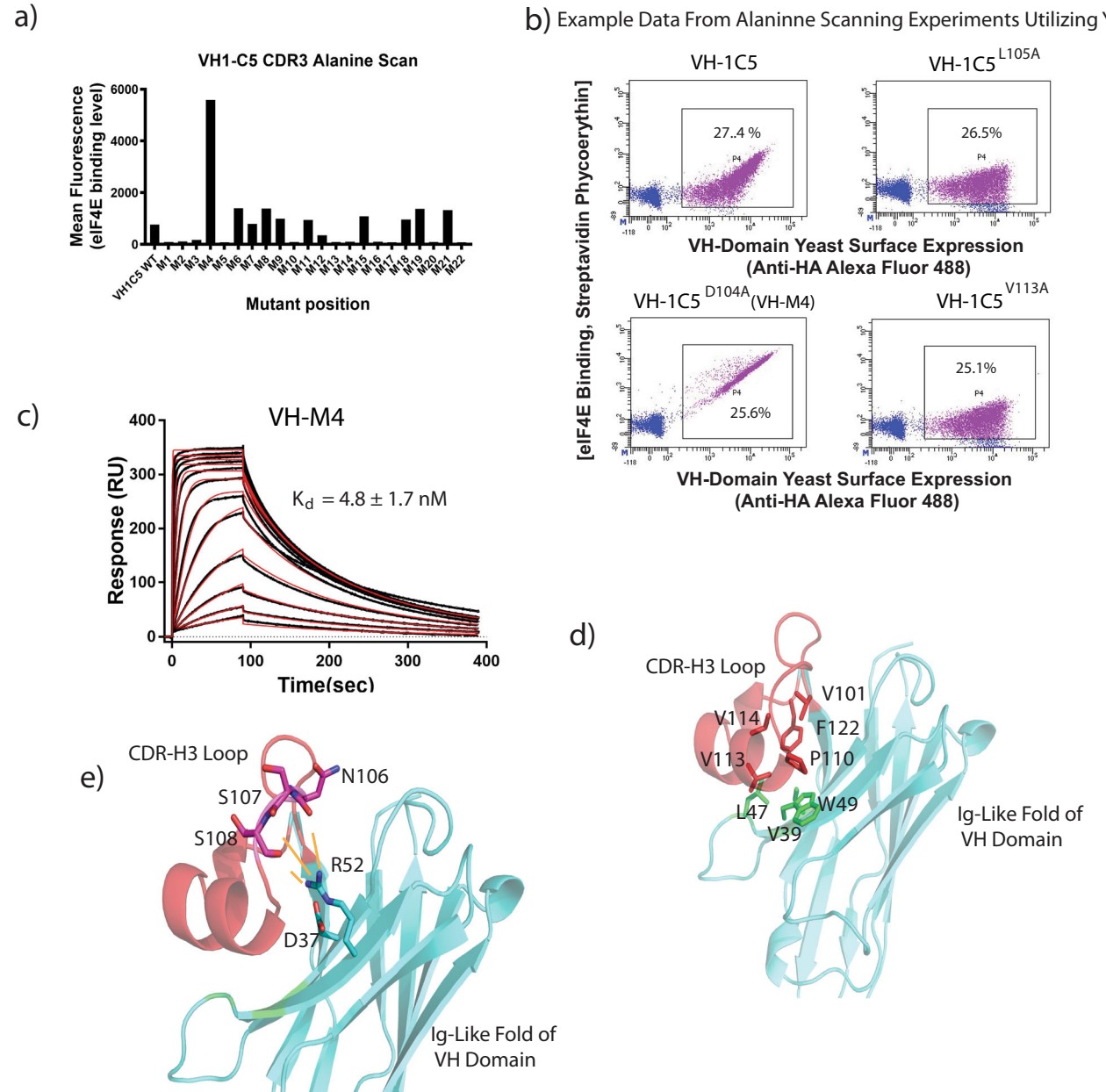

**Fig. 3 | Alanine scanning identifies an improved VH domain interactor and delineates VH domain residues distal to the interaction interface that are critical for binding. a** Scanning alanine mutagenesis was performed on the CDR3 loop of 1C5 using yeast display (YSD). Alanine mutations were made in individual yeast clones and tested for binding against fluorescently labelled eIF4E (see Fig. S15 for sorting strategy). Source data is provided in the Source Data File. **b** Example data from YSD alanine scanning experiments compared to VH-1C5. Mutations VH-1C5 [L105A] and VH1C5 L105A [V113A] demonstrate experiments where binding to eIF4E was lost, whilst VH-1C5 [D104A] shows an experiment where the alanine substitution resulted in increased binding to eIF4E. **c** Surface plasmon resonance sensogram of VH-1C5[E104D] being titrated against eIF4E immobilised via amine coupling on a CM5

chip (see materials and methods), using a twofold concentration series from 2.5 μM to 2.5 nM. $K_d$ value shown as mean ± standard error and was calculated from $n = 2$ independent experiments. **d** The eIF4E interacting CDR3 loop is highlighted in red. Residues shown in stick representation highlight critical amino acids that when mutated disrupt the hydrophobic cluster that stabilises the fold of the CDR3 loop (V101, V113, V114, P110 and F122), which is critical for binding eIF4E. Residues highlighted in green depict amino acids that contribute to the hydrophobic cluster that are located in the main fold of the VH domain (L47, V39 and W49). **e** The CDR3 loop is further stabilised by electrostatic interactions between R52 with the backbone carbonyls of N106, S107 and S108. Residue D32 orientates R152 into an optimal position to stabilise the CDR3 loop by forming a salt-bridge with it.

demonstrating the specificity of the VH domains for eIF4E (Fig. 5a). VH-S4's ability to interact with eIF4E was also compared to a constitutively active form of the repressor protein 4E-BP1 (termed 4E-BP1[4AlA]), where the phosphorylation sites (Thr37, Thr46, Ser65, Thr70)[30] responsible for modulating its binding with eIF4E were mutated to alanine. The $K_d$ of 4E-BP1[4ALA] was determined to be 0.68 ± 0.13 nM, ~10-fold weaker than the VH-S4:eIF4E interaction, with a substantially

weaker off-rate (Table 1 and Fig. S8). Both constructs exhibited comparable levels of activity in immuno-precipitating eIF4E, whilst the 4E-BP1[4ALA] negative binding control (termed 4E-BP1[YLM]) had negligible effects. VH-S4 also more efficiently disrupted the eIF4G-eIF4E interaction in cells than the weaker affinity VH domain variants (1C5, S2 and M4) using the NanoBit eIF4E:eIF4G[606–646] cell-based assay[41] (Fig. 5b). 4E-BP1[4ALA] was also assessed and again was similar in activity to VH-S4. All

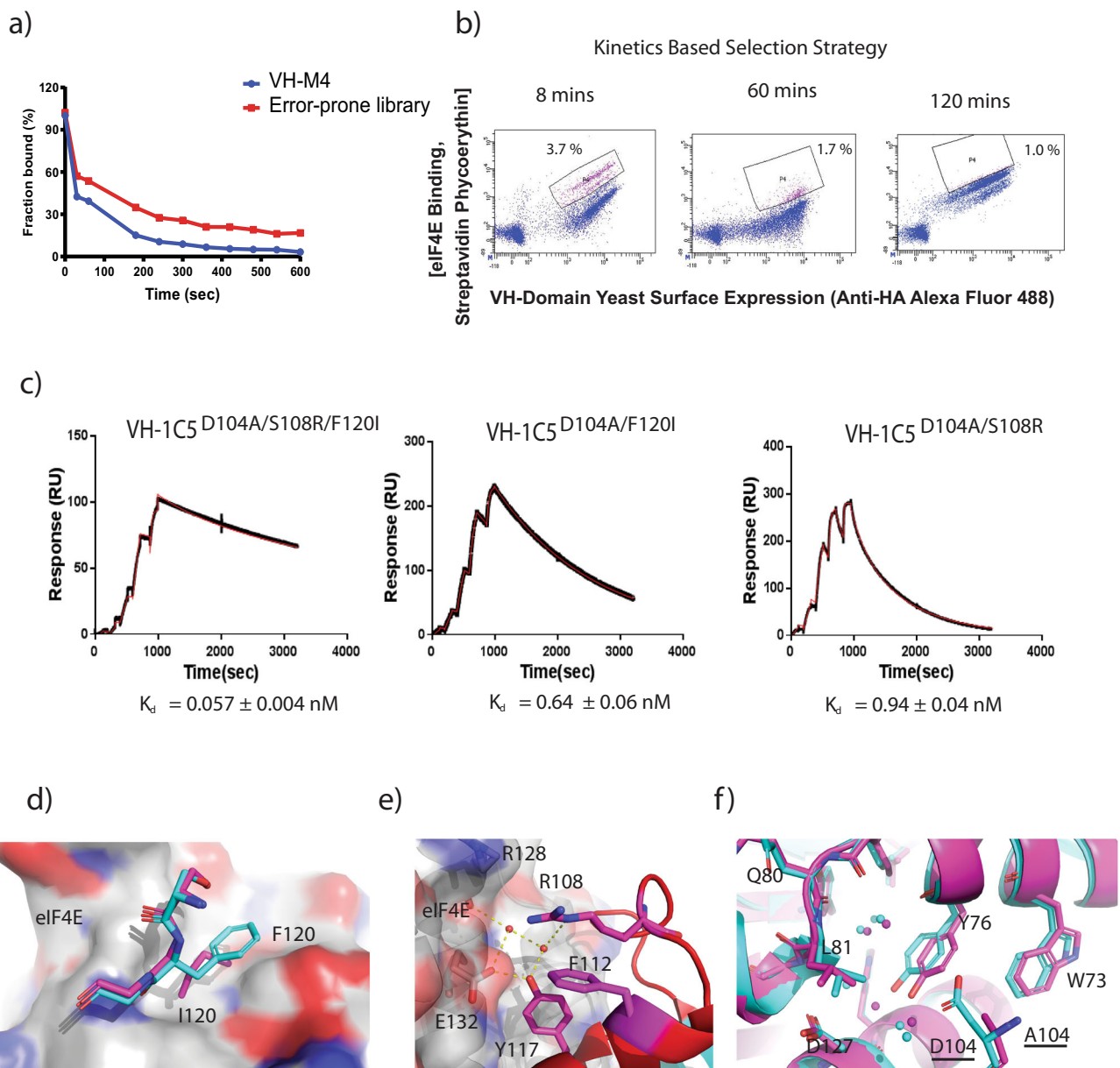

**Fig. 4 | VH-1C5 D104A domain affinity maturation using yeast surface display (YSD). a** Dissociation rate of VH-1C5D104A was compared to the error prone library, which was generated using VH-1C5D104A as template, to determine optimal competition time for dissociation rate engineering. Optimal competition time was approx. 8 min. **b** High affinity VH-1C5D104A variants were selected via three rounds of kinetic selection. Initial selectants were isolated using flow cytometry after competition with excess non-fluorescently labelled ligand after 8 mins. Selectants were then amplified and then kinetically selected after 60 mins. This was then repeated a third time with an incubation period of 110 mins eIF4E (see Fig. S15 for sorting strategy, % of cells selected for each round indicated). **c** Single representative injection cycle surface plasmon resonance sensograms of VH1C5D104A/S108R/F120I, VH-1C5D104A/F120I and VH-1C5D104A/S108R being titrated across eIF4E immobilised via amine coupling on CM5 chips, using threefold concentration series from 0.37 nM to 30 nM (see "Methods"). Binding and kinetic data (mean ± standard error) are shown on the graphs and were calculated from $n = 2$ independent experiments. **d** Mutation of F120I that occurs in the high-affinity VH-1C5D104A/S108R/F120I domain results in improved surface recognition of eIF4E. **e** The arginine that replaces S108 results in several new interactions. It engages a structured water network that forms hydrogen bonds with the backbone carbonyl of R128 and with both E132 and Y117. In addition, it stabilises the fold of the CDR3 loop further by forming a cation-π interaction with F112. **f** Overlay of VH-1C5:eIF4E (cyan) and VH-1C5D104A/S108R/F120I:eIF4E (magenta) complex crystal structures showing the conformation change induced in Y76 of eIF4E upon mutation of D104A. Associated structured water network also migrates with change in Y76 conformation.

the VH domains except the control were then shown to inhibit cap-dependent translation using a bicistronic reporter assay, with the magnitude of cap-independent inhibition again correlating to the affinity of the eIF4E interacting VH domains (Fig. 5c and Fig. S9). It is well established that disruption of the eIF4F complex prevents eIF4G mediated phosphorylation of eIF4E by Mnk1[42]. Therefore, in parallel, lysates were co-prepared from the cells used in the NanoBit assay and the levels of phosphorylated eIF4E detected (Fig. 5c). eIF4E phosphorylation as expected closely followed the levels of eIF4F complex

disruption measured in the NanoBit eIF4E:eIF4G606–646 system. 4E-BP14ALA also specifically decreased eIF4F complex formation and phosphorylation levels in a similar manner to the VH-S4 domain (Fig. 5c).

The effects of eIF4F complex disruption directly upon cap-dependent protein translation were monitored by measuring Cyclin D1 protein levels[7,43]. Increasing amounts of VH-M4 and VH-S4 expression plasmid were transfected into mammalian cells with a concomitant inhibitory effect on eIF4E phosphorylation and cyclin D1 protein levels (Fig. 5d). This effect of eIF4F disruption upon

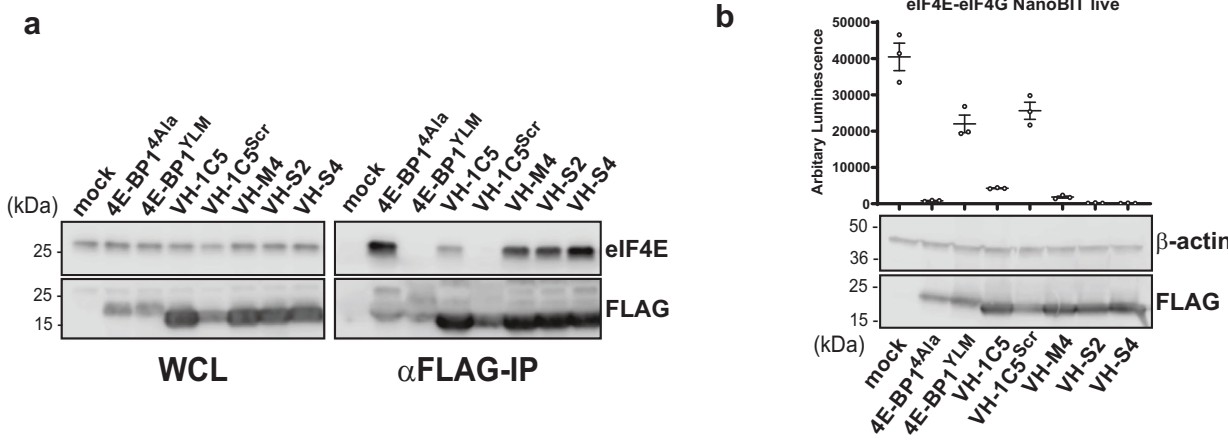

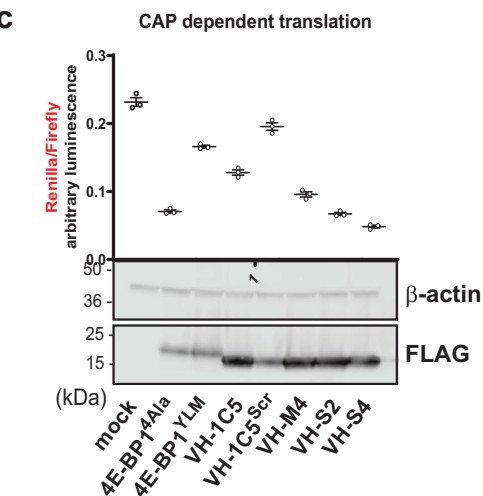

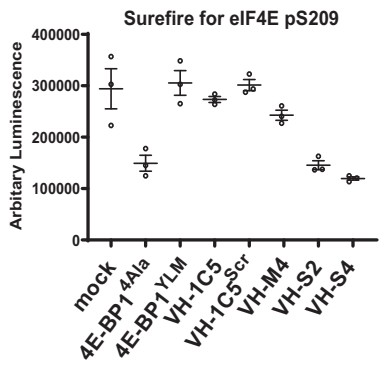

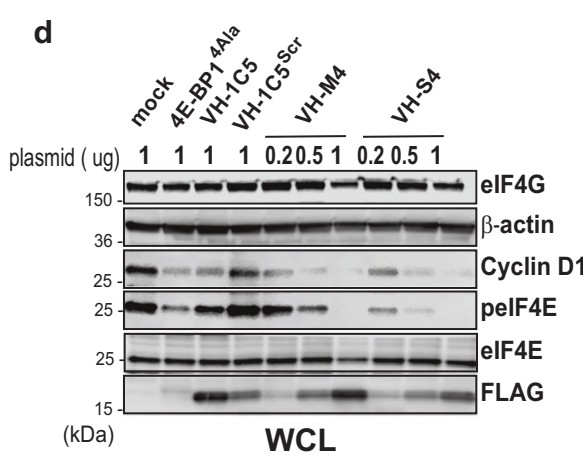

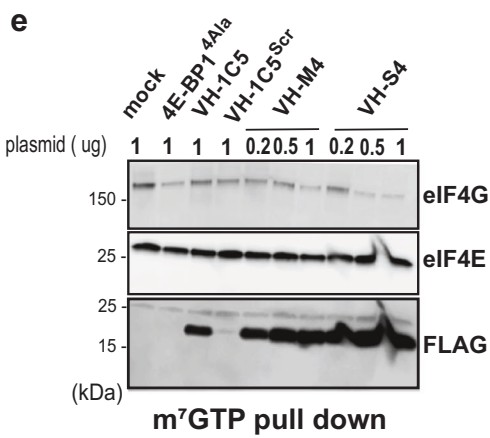

*CCND1* translation and eIF4E phosphorylation was also reflected by transfection experiments with 4E-BP1⁴ᴬᴸᴬ (Fig. 5d). m⁷GTP mediated pull down experiments verified that all the VH domains (S4, M4, S2 and 1C5) and 4E-BP1⁴ᴬᴸᴬ, except for the controls, were able to competitively displace eIF4G from eIF4E and decrease the amount of endogenous eIF4F complex detected (Fig. 5e). In addition, the specificity of VH-S4 for eIF4E was demonstrated through anti-FLAG mediated IP of VH-S4 from whole cell lysate, where no other

members of the eIF4F complex were detected apart from eIF4E (Fig. S10).

## VH-S4 modulates eIF4F mediated signalling pathways specifically

eIF4E inhibition reduces the expression of malignancy-related proteins (e.g., cyclin D1[44], Mcl-1[45], and BCL-xl[46–48]). In addition, eIF4E inhibition causes significant reductions in cellular proliferation and induction of

**Fig. 5 | Intracellular VH Domain expression disrupts eIF4F complex formation and cap-dependent translation in vitro. a** Western blot analysis of 293FT cells transfected for 48 h with FLAG tagged empty vector (Mock), 4EBP1 and anti-eIF4e VH mutants, in whole cell lysate (left, WCL) and FLAG immunoprecipitation analysis (right). **b** The NanoBit assay consists of a split luciferase fused either to eIF4E or eIF4G[606-646], whose interaction restores luciferase function[41,74]. Exogenous expression of antagonists of the eIF4E:4G interaction result in a decrease in total luminescence. SmBiT-eIF4E and eIF4G[604-646]-LgBiT constructs were co-transfected into HEK293 cells with either empty vector (Mock) or vectors containing 4EBP1 or VH mutants (as indicated) for 48 hrs. After 4 h of incubation, the luciferase activities of the reconstituted NanoBit protein were measured in living cells as described in Material and Methods. All values represent mean ± standard deviation ($n = 3$, independent experiments). **c** A bicistronic luciferase reporter (left panel), which measures the relative amount of cap-dependent translation (Renilla) to cap-independent translation (Firefly), was co-transfected with indicated plasmid. Renilla and Firefly luciferase activity was measured 48 h post transfection and plotted as a ratio-metric value. The Renilla luciferase is controlled by a GC rich

5'UTR whose activity correlates to cap dependent translation, whilst the Firefly luciferase is regulated by an internal ribosomal site (IRES). Lysates were prepared from the transfected cells and blotted for the presence of the respective constructs (below left-hand graph). In the right-hand graph, the lysates from the bicistronic assay were probed for eIF4E serine 209 phosphorylation eIF4E with the Alpha Surefire assay (Perkin-Elmer). This assay is a no wash sandwich bead-based immunoassay that enables quantification of eIF4E phosphorylation levels. All values represent mean ± standard deviation ($n = 3$, independent experiments). **d** Hela cells were transfected with indicated DNA using different amounts (μg) for 48 h and levels of indicated protein measured with western blot. Protein loading was assessed with β-actin antibody. E) Lysates from the transfected HELA cells in **d** were also used to perform m⁷GTP pull down followed by western blot analysis. Levels of 4E-BP1[4ala] and VH domain mutants were detected using anti-FLAG. All western blot analyses were repeated twice to verify their reproducibility. Representative blots are shown. Source data for **a**, **b**, **c**, **d** and **e** are provided in the Source Data file. Center line and limits denoted on dot blots represent the mean value and standard deviation of the replicates, respectively.

apoptosis. To examine the effects of VH-S4 and 4E-BP1[4ALA] mediated inhibition of eIF4E, stably transfected inducible expression systems were constructed for both proteins in A375 melanoma and MBA-MD-321 breast carcinoma cells. VH-S4 and 4EBP1[4Ala] were both induced with doxycycline for 24 h in both cell lines and were shown to decrease eIF4F complex formation (Fig. S11A) in comparison to mock control cells. Both proteins' expression was induced for over 7 days and also caused significant decreases in the cellular proliferation and viability (Fig. 6a, b, Fig. S11C and D). The most dramatic effects were seen in A375 cells, where both 4E-BP1[4ALA] and VH-S4 decreased cellular proliferation and viability to the same extent (Fig. 6a, b). In contrast, VH-S4 was less efficacious than 4E-BP1[4ALA] in MD-MBA-231 cells, despite disrupting eIF4F complex formation to the same extent (Fig. S11B).

Both VH-S4 and 4E-BP1[4ALA] when induced for 24 h resulted in significant decreases in the protein expression levels of cyclin D1[44] and of MCL-1[45], but had little effect on Bcl-xl[46-48] levels in A375 cells (Fig. 6c). Similar effects on Mcl-1 and Cyclin D1 levels were observed in the MDA-MB-231 cell line (Fig. S11B). A375 cells demonstrated no evidence of apoptosis as measured by PARP cleavage with the induction of either miniprotein, post 24 and 72 h (Fig. 6c). This result suggests that the down-regulation of Mcl-1 is insufficient to induce apoptosis in A375 cells and that the decrease in cellular proliferation measured with both VH-S4 and 4E-BP1[4ALA] is principally driven by the reduction in Cyclin-D1 protein levels (Fig. 6c). In addition, both VH-S4 and 4E-BP1[4ALA] reduced the total protein expression of 4E-BP2 with negligible effects on 4E-BP1 levels. Protein expression levels was reassessed post 72 h doxycycline induction, which revealed that Mcl-1 was no longer significantly repressed but showed that 4E-BP1 levels were reduced with a concomitant decrease in its phosphorylated forms (Fig. S12A). This lack of sustained decrease in Mcl-1 levels offers further explanation of the absence of apoptosis with eIF4F complex disruption. In parallel experiments, both VH domains and 4E-BP1[4ALA] inhibited global protein synthesis by approximately 50% as determined using puromycin pulse chase experiments (Fig. S12B). Results that correlated with the significant effects of eIF4F complex disruption upon cellular proliferation (Fig. 6a). We also verified the role of eIF4F in mediating STAT1 levels in IFN-γ treated A375 cells (Fig. 6d)[49], where selective induction of VH-S4 or 4E-BP1[4ALA] in IFN-γ treated cells reduced STAT1 protein levels.

VH-S4 specificity was further assessed by examining its effects on the pathways (AKT/mTORC and RAS/ERK) that regulate the eIF4F complex. Many of the chemical reagents used to study the biological function of the eIF4F complex modulate these pathways, and as a result elicit eIF4F independent effects e.g. PP242[50], an ATP competitive inhibitor of mTORC1/2 that leads to dephosphorylation of both its downstream targets, 4E-BP1 and S6 kinase. A375 cells were treated with either PP242 or staurosporine (a non-selective kinase inhibitor)[51] and their effects on eIF4E, AKT, ERK and rpS6 phosphorylation compared to

disruption of the eIF4F complex by both 4E-BP1[4ALA] and VH-S4 (Fig. 6c). Both miniproteins, caused dephosphorylation of eIF4E in a manner similar to PP242 through inhibition of eIF4G mediated MNK1 phosphorylation[41]. As expected, PP242 reduced 4E-BP1 phosphorylation to mediate its known effects on eIF4F complex disruption[50]. However, PP242 unlike the miniproteins also reduced phosphorylation of rS6 and AKT phosphorylation though its effects on mTORC1/2 (Fig. 6c)[50,52]. A375 cells treated with 0.2 μM of the broad kinase inhibitor staurosporine resulted only in the dephosphorylation of S6 kinase. Neither the miniproteins nor the small molecules tested affected ERK phosphorylation.

The effects of small molecules upon the eIF4F complex were extended to a wider set of compounds (including the eIF4E inhibitor, 4EGI-1 and the Mnk Kinase inhibitor, CPG-57380) and assessed at 72 h (Fig. 6e). The effects of PP242 and staurosporine on protein expression levels of Mcl-1 at 24 h (Fig. 6c) were alleviated by 72 h (Fig. 6e), whilst maintained with respect to cyclin-D1, 4E-BP1 and 4E-BP2. CPG-57380 as expected resulted in dephosphorylation of eIF4E at 72 h (Fig. 6e) with negligible effects on cyclin-D1 or 4E-BP1/2 protein levels. Additionally, PP242 and staurosporine-induced apoptosis at 72 h as indicated by PARP cleavage (Fig. 6e and Fig. S12C) in contrast to the miniproteins demonstrating that specific inhibition of the eIF4E:4G interface does not lead to apoptosis and that this is the result of other modes of action by either small molecule. Interestingly 4EGI-1, a molecule that allosterically disrupts binding of eIF4E to eIF4G, also elicits PARP cleavage and promotes eIF4E phosphorylation in contrast to VH-S4 and 4E-BP1[4ALA].

## The presence of C22 and C92 in VH-S4 is not detrimental to function or intracellular expression

With the VH-S4 domain demonstrating sufficient mammalian cellular expression to perturb eIF4F function, we assessed the effects of reintroducing C22 and C92 into VH-S4 upon its performance, whereupon negligible differences in cellular expression, proteosomal stability and in the amounts of eIF4E immune-precipitated were observed between the two variants (Fig. 7a, b). This prompted further examination of the differences in mammalian expression levels of the various VH domain mutants evolved at different stages of the directed evolution process (Fig. 7c) with or without C22 and C92. From these results, it was apparent that the 4D5 template sequence with or without C22 and C92 was suboptimal for cellular expression and that improvements in the thermal stability of the VH variants generally correlated with increased soluble mammalian cellular expression (Fig. 7c and Fig. S3). Incorporation of C22 and C92 predominately increased the thermal stability of the purified VH domains by approx. 10 °C indicative of the formation of intra-domain disulfide bond formation in vitro (Fig. S13). However, the formation of this bond in the cytoplasm is difficult to establish but it is apparent that the other mutations in VH-36 and later

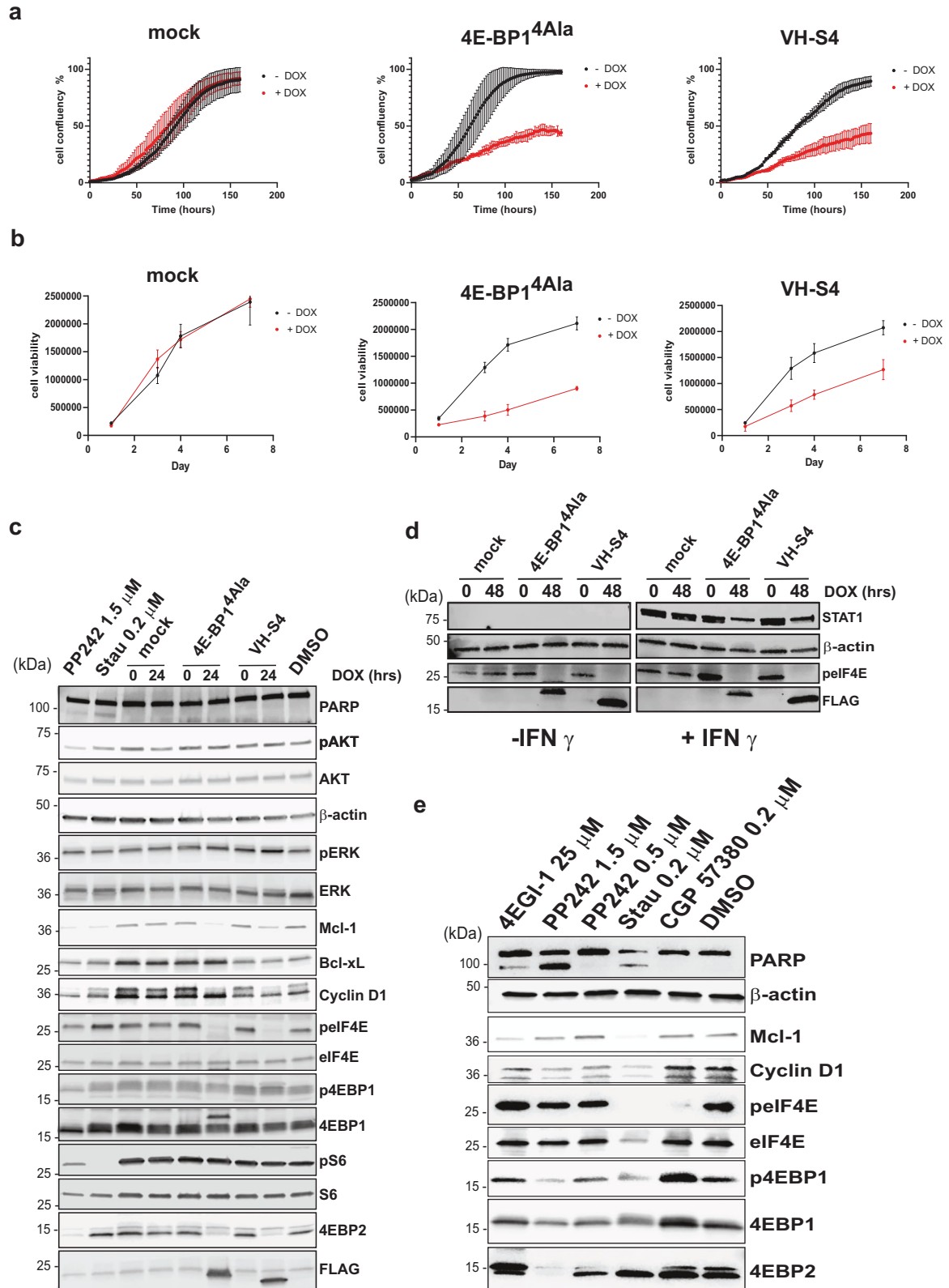

**Fig. 6 | VH-S4 modulates eIF4E-mediated signalling pathways. a** A375 Mock, 4E-BP1[4Ala] and VH-S4 inducible cell line confluence was measured in presence or absence of doxycycline for the indicated time (hrs) using an Incucyte (EssenBiosciences). **b** A375 stable cell lines treated as in A) were assayed for viability over the indicated time (days). All values represent mean ± SD (n = 3). **c** Inducible A375 cells harbouring Mock, 4E-BP1[4Ala] and VH-S4 were incubated for 24 hrs with or without 1 ng/ml of Doxycycline. Lysates were analysed by western blot using the antibodies indicated in the blot (for further details see materials and methods). β-Actin was used as a loading control. **d** A375 inducible stable cell lines were simultaneously incubated with or without doxycycline and where either treated with or without IFN-γ (50 ng/ml) for 48 h. After cell lysis, extracts were analysed by western blot and protein levels for STAT1, Phosho-eIF4E, eIF4E and FLAG tagged 4E-BP1[4Ala] and VH-S4 were measured. **e** A375 cells were treated with the indicated compounds at the labelled concentrations for 72 h with a residual DMSO concentration of 1% (v/v). Lysates were analysed by western blot using the antibodies indicated in the blot (for further details see "Materials and methods"). All western blot analyses were repeated twice to verify their reproducibility. Representative blots are shown. Source data for **a**, **b**, **c**, **d** and **e** are provided in the Source Data file.

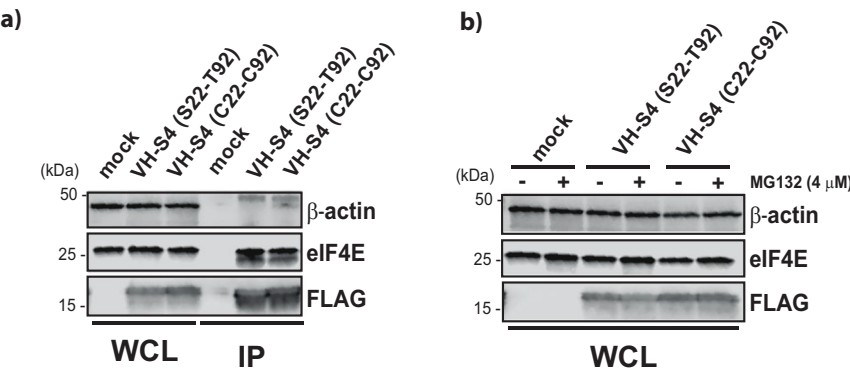

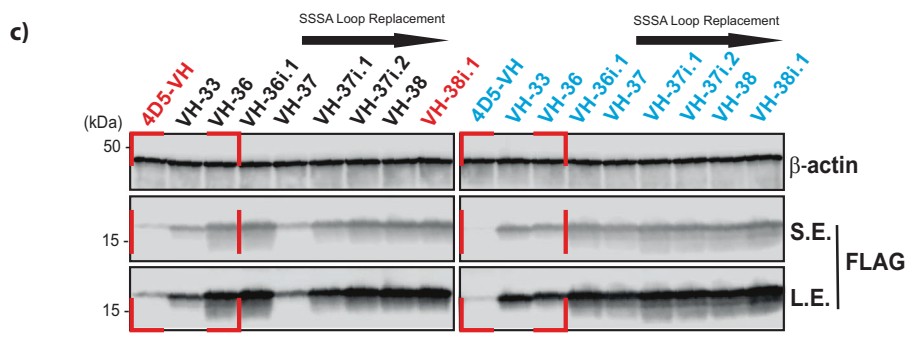

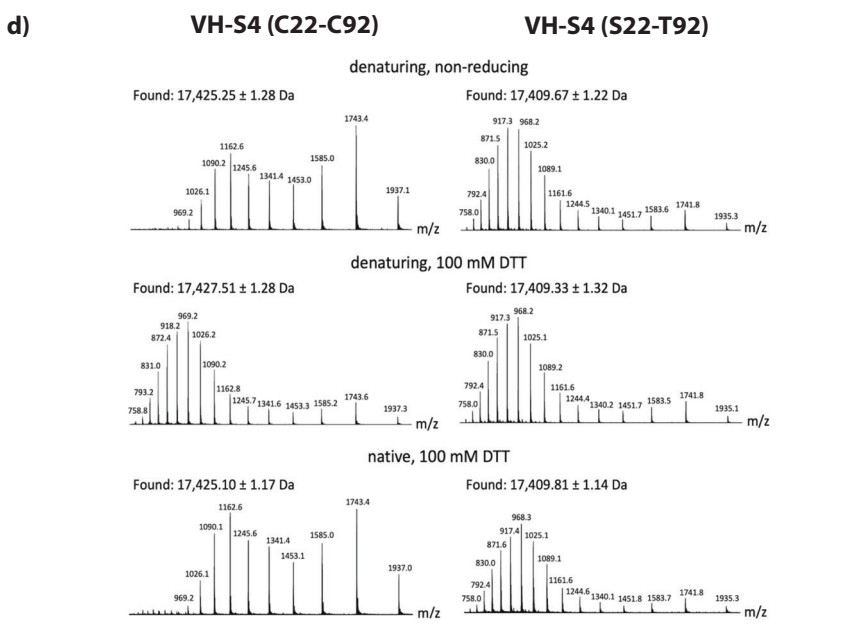

variants contributed to the soluble and stable expression of the VH domain constructs intracellularly in the absence of C22 and C92. The replacement of the previous CDR3 loop in VH-37i resulted in loss of cellular expression and thermal stability that was either compensated for by the re-introduction of C22 and C92, suggesting that these residues are forming stabilising interactions with each other inside the cell, or by the incorporation of S93V and A78V in

VH-37i.1 and VH-37i.2, respectively, that both facilitate improved scaffold stability.

**The VH-S4 intra-domain disulfide bond is retained in a reducing environment**

The rescue of intracellular expression of VH-37i by the re-insertion of C22 and C92 raises the possibility of intra-disulfide bond formation

**Fig. 7 | Intra-domain disulfide bond is not detrimental functionally to the evolved VH-S4 domain and is retained in a reducing environment. a** 293FT cells were transfected for 48 h with FLAG-tagged empty vector (Mock), VH-S4(S22C-T92C) and VH-S4. Cell lysates were then subjected to Flag immunoprecipitation (IP) followed by western blot analysis. to detect the presence of endogenous eIF4E as co-immunoprecipitant in the IP. **b** Cells transfected as in **a** were cultured in presence or absence of MG132 for 18 h. Lysates were analysed by western blot and level of endogenous eIF4E and overexpressed Flag-tagged miniproteins were assessed using an anti-eIF4E and anti-Flag antibody. **c** Western blot analysis of whole cell lysates from 293FT cells transfected for 48 h with FLAG-tagged directed evolution VH mutants, mock and corresponding controls harbouring relevant amino acid replacements at position 22 and 92 as denoted. Removal of the W[95]GGDGFYP[100B] loop with SAAA is indicated. Cellular expression of VH domain proteins was

assessed with anti-flag antibody over a short (S.E.) and long western blot exposure (L.E.) β-actin was used as loading control for all the experiment. **d** Recombinant VH-S4(S22C-T92C) and VH-S4 were analysed by liquid chromatography/mass spectrometry, after treatment with the indicated conditions. Mass spectra were integrated across the principle components of the resulting total ion chromatograms, and the experimental masses were taken as the average across individual charge state ions. The deconvoluted masses have experimental errors of ~300 ppm (based on molecular weights of 17,416.15 Da and 17,434.25 Da for VH-S4 and the reduced form of VH-S4(S22C-T92C), respectively), which are of the order expected for a single-quadrupole mass analyser. All western blot analyses were repeated twice to verify their reproducibility. Representative blots are shown. Source data for **a** and **b** are provided in the Source Data file.

occurring inside the cell, especially with both residues residing internally near each other in the VH domain and occluded from the reducing cellular environment. To establish the presence a disulfide, recombinant VH-S4(S22C-T92C) was analysed by liquid chromatography/mass spectrometry after treatment with defined redox conditions (Fig. 7d). Treatment of VH-S4(S22C-T92C) with the reducing agent DTT and the chemical denaturant guanidinium chloride led to a molecular weight shift of approximately 2 Da, consistent with the loss of an intra-domain disulfide bond. As expected from literature precedent[53,54], disulfide reduction was accompanied by a shift in the spectral envelope toward higher charge states, providing a second line of evidence that a disulfide had been present before reducing/denaturing treatment. No significant shift in mass or spectral envelope was measured after treatment with DTT in non-denaturing conditions, suggesting the disulfide would be stable intracellularly. Together this data suggests that recombinant VH-S4(S22C-T92C) contains a disulfide bond that is preserved in the reducing environment of the cell. VH-S4(S22C-T92C) was also crystallised in complex with eIF4E, where the presence of the intra-domain disulfide bonds caused no conformational changes in the global fold of the VH domain compared to the disulfide-free domain (Fig. S14). Interestingly, the S108R sidechain is found forming a direct electrostatic interaction with E132 and displacing the structured waters observed in the VH-S4 structure (Fig. 4d and Fig. S14).

## Discussion

We have developed a repertoire of VH domain-based miniproteins that sample a wide range of $K_d$s (picomolar to micromolar), which disrupt the eIF4F complex though a distinct binding pose and that are amenable to cellular expression. We have also established that the residues (C22-C92) required for intra-domain disulfide bond formation are not detrimental to the expression of the majority of VH variants with improved thermal stabilities (Fig. 7c) and that these residues form a disulfide bond when re-substituted into the VH-S4 domain in the presence of a reducing environment. The re-introduction of the intra-domain disulfide bond also has no effect on the VH domain structure or its interaction with eIF4E (Fig. 7 and Fig. S14). Further, the re-substitution of C22 and C92 into VH-37i restored its cellular expression in the absence of other VH variant mutations (VH-37i.1 and VH-37i.2) that thermally stabilized the VH domain (Fig. 7c). These results suggest that if poorly expressing VH domains transiently form correctly folded structures upon protein synthesis, then occlusion of C22 and C92 from the reducing environment will enable disulfide formation and allow improved cellular expression. This suggests that in phage selections utilizing the disulfide-free VH scaffold, which fail to identify a clone amenable to cellular expression, re-substitution of C22 and C92 is a potential rescue strategy.

In contrast to PPIs such as Mdm2:p53[55] and Bcl-2:Bax[56], the eIF4E:4G[34] interaction interface is relatively planar with no distinguishing clefts and is much more structurally similar to PPIs such as the interface found in the Notch transcription complex between ICN:CSL and MAML and β-catenin:TCF[57]. The eIF4E interacting VH-S4 domain

has evolved an optimised CDR3 loop for recognizing planar PPI surfaces, and with randomization of the appropriate residues should serve as an ideal template for building libraries to explore these molecular surfaces. In addition, no clinically approved drugs exist that target the eIF4E:4G interface directly, despite the description of several antagonistic small molecules that albeit interact weakly with eIF4E and have little cellular efficacy[2]. The VH-S4/1C5 structures offer two possible avenues for future lead molecule development: (1) the CDR3 loop could be used as a template to drive forward a macrocyclic discovery strategy and (2) the critical residues involved in binding eIF4E could be used as a basis for small molecule design.

The applicability of VH domains for cellular pathway modulation and therapeutic modelling were demonstrated by comparing their mode of action to several small molecules and a phosphorylation deficient mutant of 4E-BP1 (4E-BP1[4ALA]) known to modulate the eIF4F complex. Both 4E-BP1[4ALA] and VH-S4 exhibited similar potencies to each other in terms of their bioactivity, which is unexpected as VH-S4 interacts more strongly with eIF4E (Table 1). 4E-BP1[4ALA] was also more effective in reducing cellular proliferation of MB-MDA-231 cells despite similar efficacies to VH-S4 in terms of eIF4F complex disruption. These results suggest that other mechanisms apart from displacement of eIF4G are putatively playing a role in the effects of 4E-BP1[4ALA] mediated inhibition. Such effects could be post-translational modifications apart from phosphorylation (e.g., ubiquitination) or additional inhibition by endogenous 4E-BP1 through unappreciated feedback loops. Therefore, VH domains that are inert to cellular regulation pathways can be used to decipher their effects on naturally occurring repressors provided they interact at the same binding site.

Furthermore, the VH-S4 domain was compared to several well-known compounds used to study the role of the eIF4F complex. For example, inhibitors of mTORC1, the multi-subunit protein responsible for phosphorylating 4E-BP1 and controlling its ability to inhibit eIF4E, is also responsible for regulating the activity of other target proteins and their downstream pathways, such as the rS6 kinase[58,59]. Here we demonstrated that VH-S4 disrupts eIF4F complex disruption in the absence of effects on rS6 kinase phosphorylation (including 4EBP1 also) and is thus a more appropriate tool for understanding the impact of eIF4E. Additionally, we also compared it to the small molecule inhibitor of the eIF4E:4G interaction, 4EGI-1[35], where in contrast to the VH-S4 domain the induction of apoptosis was seen. A result shared with the in mTORC1 inhibitor that was used (PP242). These results further demonstrate how specific miniproteins can be used to refine our understanding of small molecule inhibition on cellular phenotypes, and to delineate whether desirable outcomes such as apoptosis are the result of direct on target engagement or indirect off-target effects. Tools that could be useful for therapeutic design and research.

## Methods

### Library generation for stability improvement
Random mutagenesis libraries were created by error-prone PCR using Genemorph II Mutagenesis Kit (Stratagene). The cloned wild-type

genes (for first library) or improved variants (for subsequent libraries) were used as templates, and the reactions were performed according to the manufacturer's protocol. The primers used annealed to sequences flanking the open reading frame (ORF) to be mutated. An average of 50 ng of insert template was used, in a 30 cycles PCR reaction with phosphorylated primers, leading to an average of ~2–3 amino-acid mutations per gene in the final plasmid library. The error-prone PCR product was gel purified using a Qiagen Gel purification Kit.

The plasmid library was created by Megaprimer PCR of Whole Plasmid (MEGAWHOP) reaction[60]. Typically, 1–10 ng of plasmid containing the wild-type (WT) gene was used as template, and amplified using approximately 1 ng of insert per bp of insert length of purified error prone PCR product as the megaprimer (e.g., 500 ng for a 500 bp insert), in a PCR reaction with KOD Xtreme polymerase (Merck). The buffer composition was the standard recommended by the manufacturer, with the addition of 1 mM NAD+, 40 U of Taq DNA ligase, and using 0.5 U of KOD Xtreme polymerase. The PCR conditions were as follows: 94 °C for 2 min, (98 °C for 10 s, 65 °C 30 s, 68 °C for 6 min) for 30 cycles, 4 °C on hold. After PCR, 20 U of DpnI (NEB) was added to the PCR reaction, and incubated for 3 h at 37 °C.

To generate a library that would allow the identification of a disulfide-free version of VH36, first the stII signal sequence was removed by site-directed mutagenesis, generating clone VH36i. Then, a site directed mutagenesis library was generated by MEGAWHOP, in which residue positions C22, A24 and C96 were randomized. A PCR using phosphorylated forward (5′-GGCTCACTCCGTTTGTCCNNKG-CANNKTCTGGCTTCAACATTAAAGAC-3′) and reverse (5′-CCTCCCCA GCGGCCMNN ATAATAGACGGCAGTG-3′) primers was performed using VH-36i as template, and KOD HotStart polymerase (Merck), according to the manufacturer's protocol. The PCR product was gel purified, and the MEGAWHOP reaction performed as described previously, after which DpnI treatment was performed. DH10B cells were electroporated with the DpnI-treated, purified MEGAWHOP product yielding libraries of ~10^5 unique members.

## Solubility and stability screening using colony filtration (CoFi)

Rosetta2 cells were transformed with the mutagenesis libraries and plated on 24.5 cm diameter square LB-Agar plates supplemented with 50 μg/mL kanamycin and 34 μg/mL chloramphenicol. These were termed the master plates. Colonies were transferred to a Durapore 0.45 μm filter membrane (Millipore), and placed on LB-Agar plates (colonies facing up) supplemented with antibiotics and 30 μM Isopropyl β-D-1-thiogalactopyranoside (IPTG). Induction was performed overnight at room temperature (RT) for protein expression.

After protein production, the induction plates were subjected to the desired temperature (room temperature (RT) for solubility screen (termed CoFi), or higher temperatures for the stability screen (termed Hot-CoFi) for 30 min. The Durapore membrane was transferred to a lysis sandwich composed of a Whatman paper, soaked in CoFi lysis buffer [20 mM Tris, pH 8.0, 100 mM NaCl, 0.2 mg/ml lysozyme, 11.2 U/mL Benzonase Endonuclease (Merck) and 1:1000 dilution of Protease Inhibitor Cocktail Set III, EDTA-Free (Merck)], a nitrocellulose membrane (Millipore) and incubated at the screening temperature for another 30 min. Cell lysis was further improved by three freeze-thaw cycles at −80 °C and RT, respectively (30 min each). The Durapore membrane and the Whatman paper were discarded, and the nitro-cellulose membrane was incubated in blocking buffer [TBS-T buffer (20 mM Tris, pH 7.5, 500 mM NaCl, 0.05% (vol/vol) Tween 20) and 1% Bovine Serum Albumin (BSA)] for 1 h at RT.

After blocking, the nitrocellulose membrane was washed three times in TBS-T, 10 min at room temperature, with shaking at 70 revolutions per minute (rpm). The presence of soluble protein was detected either by incubating the membrane in TBS-T containing a 1:5,000 dilution of HisProbe-HRP (Thermo Scientific, Cat #: 15165), or with a 1:10,000 dilution of protein A-HRP probe (Life Technologies, Cat #:

200-103-077) in TBS-T with 1% BSA, for 1 h at RT, with shaking at 30 rpm. Three washing were then performed as previously described. The nitrocellulose membrane was developed using Super Signal West Dura chemiluminescence kit (Thermo Scientific) and imaged using a CCD camera (Fujifilm LAS-4000).

The sequence of the variants producing high CoFi or Hot-CoFi signal was determined by aligning the developed images with the master plates, and picking the colonies linked to the high signals obtained. The picked colonies were grown overnight at 37 °C in LB media with 50 μg/mL kanamycin and 34 μg/mL chloramphenicol, plasmids isolated and submitted to Sanger sequencing.

## Determination of temperature of cellular aggregation (T_CAGG)

Starter cultures composed of 1 mL Terrific Broth (TB) supplemented with antibiotics were inoculated from glycerol stocks of the clones to test and incubated overnight at 37 °C with shaking. The next day, 200 μL were used to inoculate 20 mL of TB supplemented with antibiotics and incubated at 37 °C with shaking until an $OD_{600}$ of approximately 2 was reached. The cultures were then induced with 30 μM IPTG and incubated overnight at 18 °C. The expression cultures were centrifuged at 4800 × $g$ for 15 min at 4 °C, and the pellets were resuspended in 5 mL of CoFi lysis buffer. Cell lysis was performed through three freeze-thawing cycles at −80 °C and RT, respectively. The samples were then transferred to a PCR plate in 100 μL aliquots. Each aliquot was then subjected to a different temperature, using a temperature gradient in a PCR machine, for 15 min and then cooled down to 4 °C. The heat-treated samples were centrifuged at 3200 × $g$ for 15 min at 4 °C. The supernatant was transferred to a 0.65 μm filter plate and filtered by centrifugation at 2000 × $g$ for 15 min at 4 °C. 2 μL of the filtrate was dotted onto a nitrocellulose membrane and developed using HisProbe-HRP as described. After development, the average intensity of each dot was calculated using ImageJ software (NIH) and the midpoint of the transition determined using the Boltzmann equation on Prism 8.0 (GraphPad Software).

## VH domain expression and purification during the stabilization campaign

Rosetta2 cells were transformed with the corresponding pET24a-based plasmids containing the VH domain open reading frames and plated onto LB again containing 50 μg/mL kanamycin and 34 μg/mL chloramphenicol. Single colonies were grown on 4 mL TB media supplemented with 50 μg/mL kanamycin and 34 μg/mL chloramphenicol, at 37 °C. Once $OD_{600nm} = 1.5–2$, the temperature was lowered to 20 °C, 0.5 mM IPTG was added, and the culture was grown overnight, with agitation. The cultures were pelleted by centrifugation, and resuspended in 1.3 mL VH lysis buffer (100 mM HEPES pH 8.0, 500 mM NaCl, 10 mM Imidazole, 10% glycerol, 10U Benzonase (Merck), 10% n-Dodecyl ß-D-maltoside, 1 mg/mL lysozyme, 1 mM MgSO4, 1× protease inhibitor cocktail (Nacalai Tesque)), and lysed through three freeze-thawing cycles at −80 °C and RT. The lysate was centrifuged at 3320 × $g$ for 20 min at 4 °C, and the supernatant was transferred onto a purification column containing 200 μL of Ni-NTA resin (Thermo Fisher), and incubated for 15 min at RT. The column was washed with 10 column volumes of VH wash buffer (20 mM HEPES pH 8.0, 500 mM NaCl, 25 mM Imidazole, 10% glycerol) and eluted with 5 column volumes of VH elution buffer (20 mM HEPES pH 8.0, 500 mM NaCl, 500 mM Imidazole 10% glycerol). The samples were immediately desalted into PBS using PD MiniTrap G-25 desalting columns (Cytiva).

## Determination of melting temperature ($T_m$) using Differential Scanning Fluorimetry (DSF)

To perform Differential Scanning Fluorimetry[61], 7.5 μg of protein (see VH domain expression and purification section) was diluted into a 25 μL PBS buffer solution containing 5x SYPRO Orange fluorescent dye (Bio-Rad). Proteins were tested in triplicates, the fluorescence was

**Table 2 | Library construction primers, where X = 0.2 G + 0.2 A + 0.5 T + 0.1 C, Y = 0.4 A + 0.2 T + 0.4 C and Z = 0.1 G + 0.9 C**

| Primer name | Primer sequence |
|---|---|
| H1 | CCTCTGCAATTTCTGGCTTC [XYZ] NTT [XYZ][XYZ] ACT [XYZ] ATAGACTGGGTGCGTCAGG |
| H2 | CTGGAATGGGTTGCAAGGATT [XYZ] CCT [XYZ][XYZ] GGT [XYZ] ACT [XYZ] TATGCCGATAGCGTCAAGGG |
| H3.6 | GCCGTCTATTATACTGKCCGC [XYZ][XYZ][XYZ][XYZ][XYZ][XYZ] GSTNTKGACTACCGGGGTCAAG |
| H3.7 | GCCGTCTATTATACTGKCCGC [XYZ][XYZ][XYZ][XYZ][XYZ][XYZ][XYZ] GSTNTKGACTACCGGGGTCAAG |
| H3.8 | GCCGTCTATTATACTGKCCGC [XYZ][XYZ][XYZ][XYZ][XYZ][XYZ][XYZ][XYZ] GSTNTKGACTACCGGGGTCAAG |
| H3.9 | GCCGTCTATTATACTGKCCGC [XYZ][XYZ][XYZ][XYZ][XYZ][XYZ][XYZ][XYZ][XYZ] GSTNTKGACTACCGGGGTCAAG |
| H3.10 | GCCGTCTATTATACTGKCCGC [XYZ][XYZ][XYZ][XYZ][XYZ][XYZ][XYZ][XYZ][XYZ][XYZ] GSTNTKGACTACCGGGGTCAAG |
| H3.11 | GCCGTCTATTATACTGKCCGC [XYZ][XYZ][XYZ][XYZ][XYZ][XYZ][XYZ][XYZ][XYZ][XYZ][XYZ] GSTNTKGACTACCGGGGTCAAG |
| H3.12 | GCCGTCTATTATACTGKCCGC [XYZ][XYZ][XYZ][XYZ][XYZ][XYZ][XYZ][XYZ][XYZ][XYZ][XYZ][XYZ] GSTNTKGACTACCGGGGTCAAG |
| H3.13 | GCCGTCTATTATACTGKCCGC [XYZ][XYZ][XYZ][XYZ][XYZ][XYZ][XYZ][XYZ][XYZ][XYZ][XYZ][XYZ][XYZ] GSTNTKGACTACCGGGGTCAAG |
| H3.14 | GCCGTCTATTATACTGKCCGC [XYZ][XYZ][XYZ][XYZ][XYZ][XYZ][XYZ][XYZ][XYZ][XYZ][XYZ][XYZ][XYZ][XYZ] GSTNTKGACTACCGGGGTCAAG |
| H3.15 | GCCGTCTATTATACTGKCCGC [XYZ][XYZ][XYZ][XYZ][XYZ][XYZ][XYZ][XYZ][XYZ][XYZ][XYZ][XYZ][XYZ][XYZ][XYZ] GSTNTKGACTACCGGGGTCAAG |
| H3.16 | GCCGTCTATTATACTGKCCGC [XYZ][XYZ][XYZ][XYZ][XYZ][XYZ][XYZ][XYZ][XYZ][XYZ][XYZ][XYZ][XYZ][XYZ][XYZ][XYZ] GSTNTKGACTACCGGGGTCAAG |
| H3.17 | GCCGTCTATTATACTGKCCGC [XYZ][XYZ][XYZ][XYZ][XYZ][XYZ][XYZ][XYZ][XYZ][XYZ][XYZ][XYZ][XYZ][XYZ][XYZ][XYZ][XYZ] GSTNTKGACTACCGGGGTCAAG |
| H3.18 | GCCGTCTATTATACTGKCCGC [XYZ][XYZ][XYZ][XYZ][XYZ][XYZ][XYZ][XYZ][XYZ][XYZ][XYZ][XYZ][XYZ][XYZ][XYZ][XYZ] [XYZ] GSTNTKGACTACCGGGGTCAAG |
| H3.19 | GCCGTCTATTATACTGKCCGC [XYZ][XYZ][XYZ][XYZ][XYZ][XYZ][XYZ][XYZ][XYZ][XYZ][XYZ][XYZ][XYZ][XYZ][XYZ][XYZ] [XYZ][XYZ] GSTNTKGACTACCGGGGTCAAG |
| H3.20 | GCCGTCTATTATACTGKCCGC [XYZ][XYZ][XYZ][XYZ][XYZ][XYZ][XYZ][XYZ][XYZ][XYZ][XYZ][XYZ][XYZ][XYZ][XYZ][XYZ] [XYZ][XYZ][XYZ] GSTNTKGACTACCGGGGTCAAG |

monitored using a 96-well Real-Time PCR detection system (iCycler iQ, from Bio-Rad), from 25 to 95 °C, with a gradual temperature increase of 1 °C every 10 s. The melting temperature was determined using the Boltzmann equation on Prism 8.0 (GraphPad Software).

**Phage display library generation**
The open reading frame coding for VH-38i.1 was synthetized with a stII signal sequence at the 5′ end and the C-terminal domain of the M13 gene 3 at the 3′ end. Positions 28-31, 33, 52a, 53-54, 56 and 58 (according to Kabat numbering) were mutated to Ser to replace 4D5's CDR1 and CDR2 with a generic sequences, while position 93 was mutated to Gly for flexibility, to form the VH template sequence used for phage display. The gene sequence was then cloned into a pPCR-Script-SK + derived vector, downstream of the lac promoter. This vector is referred to a pVH1 vector. Phage display library creation was then performing as outlined in the following steps. pVH1 vector was transformed into CJ236 cells (Lucigen) and the cells subsequently plated on 2YT-Agar plates supplemented with 50 µg/mL carbenicillin. After overnight incubation at 37 °C, colonies were picked and used to inoculate 5 mL of 2YT media supplemented with 50 µg/mL carbenicillin. The culture was then incubated at 37 °C with shaking for 6 h, after which M13K07 helper phage (NEB) was added to a final concentration of $10^{10}$ pfu/mL. After 1 h incubation at 37 °C, the culture was transferred to 25 mL 2YT supplemented with 50 µg/mL carbenicillin and 25 µg/mL kanamycin and incubated overnight at 37 °C with agitation. Phage was precipitated with 1:5 volumes of PEG/NaCl buffer (20% PEG 8000/2.5 M NaCl), followed by single-stranded DNA (ssDNA) purification using Qiaprep Spin M13 Kit (Qiagen)[62,63].

Fifteen sub-libraries were created by Kunkel mutagenesis with primers containing skewed base compositions[64,65]. See Table 2 for the phosphorylated primers used.

The phosphorylated primers (0.6 µg of each H1, H2 and H3.6 to H3.20) were annealed to 20 µg ssDNA by incubation for 2 min at 90 °C, followed by 5 min at 50 °C and then placed on ice. The complementary, mutagenized strand was synthesized by T7 DNA polymerase and ligated with T4 DNA ligase, overnight at RT. After purification, the sub-libraries were electroporated into TG1 cells (Lucigen) infected with M13KO7 helper phage. The electroporated sub-libraries were incubated overnight at 37 °C in 2YT supplemented with 50 µg/mL carbenicillin and 25 µg/mL kanamycin, with agitation. They were purified by precipitation with PEG/NaCl buffer, followed by resuspension in PBS + 50% glycerol, and stored at −20 °C until use. The library size was determined at approximately $2.87 \times 10^{10}$ unique clones, by plating serial dilutions of the electroporated products, and the quality of the library verified by sequencing 24 random clones from each individual sub-library. CDR amino-acid composition analysis was performed by next-generation sequencing (NGS) by Illumina HiSeq method (service provider: Singapore Joint Venture & Sequencing Center NovogeneAIT) followed by sequence analysis using in-house software.

**Expression, refolding, and purification of eIF4E**
Rossetta pLysS competent bacteria were transformed with the pET11d expression plasmid containing the full-length eIF4E clone. The cells expressing the full-length eIF4E construct were grown in LB medium at 37 °C to an $OD_{600}$ of ~0.6 and eIF4E induction was started with 1 mM IPTG. The culture was immediately placed in a shaker-incubator for 3 h at 37 °C. Cells were harvested by centrifugation and the cell pellets were re-suspended in 50 mM Tris pH 8.0, 10% sucrose, and were then sonicated. The sonicated sample was centrifuged for 10 min at 17,000 × g at 4 °C. The resulting pellet was re-suspended in Tris/Triton buffer (50 mM Tris pH 8.0, 2 mM EDTA, 100 mM NaCl, 0.5% Triton X-100). The sample was then centrifuged at 25,000 × g for 15 min at 4 °C and the pellet was re-suspended in Tris/Triton buffer. After re-centrifugation, the remaining pellet was solubilised in 6 M guanidinium hydrochloride, 50 mM Hepes-KOH pH 7.6, 5 mM DTT. The protein concentration of the sample was then adjusted to 1 mg/mL. The denatured protein was refolded via a 1/10 dilution into refolding buffer consisting of 20 mM Hepes-KOH, 100 mM KCl and 1 mM DTT. The refolded protein was concentrated and desalted using a Amersham PD10 column into refolding buffer. The eIF4E protein sample was run over a monoQ column and eluted with a 1 M KCl gradient. eIF4E eluted as a sharp peak at a ~0.3 M KCl.

## eIF4E biotinylation for use in phage selections

Sulfo-NHS–LC–LC biotin was added in an equimolar ratio to a solution of eIF4E at a concentration of a 100 μM and incubated at room temperature (Thermofisher Scientific). After 1 h, unreacted biotin was removed by passing the solution over a fast-desalting column (equilibrated with phosphate-buffered saline) twice. Biotinylated eIF4E was stored at 4 °C for a maximum period of up till 1 week.

## eIF4E Phage display library screening

500 μL of the phage display library (~2.5 × 10¹³ pfu) was precipitated with PEG/NaCl buffer and resuspended in BSA block buffer (BBB): 4% BSA in PBS supplemented with 0.05% Tween20 (PBST). Immunotubes (Nunc) were coated with NeutrAvidin (Thermofisher Scientific, 5 μg/ml of Neutravidin in PBS) and incubated at 4 °C overnight. The coated tubes were then washed with PBS (1x) and blocked with BBB for 1 h at RT. After washing the tube with PBST, 100 μl of biotinylated eIF4E (5 μg/ml in PBST) was then added and incubated in the tubes for 1 h at RT. A negative selection tube was also prepared as described above, but without adding eIF4e. The tubes were then washed with PBST (3×). The phage library was first incubated for 1 h at RT in the negative selection tube, then transferred to the eIF4E coated immunotube and incubated with the target protein for 1 h with rotation. The non-bound phage were then washed away with three washes of BBB, followed by three washes with PBST and then two washes with PBS. The bound phage were eluted with 1 mL of 1 mg/mL trypsin in trypsin buffer (TBS + 2 mM CaCl₂). The eluted phages were used to infect 5 mL of a TG1 culture (in 2YT media) in exponential growth phase (OD600 ~ 0.5), incubated 30 min at 37 °C. From round 2 onwards, 1.2 mL of infected TG1 cells were stored with 20% glycerol at −80 °C, to be used for monoclonal screening (glycerol stocks for monoclonal screening). The remaining infected TG1 cells were transferred to 50 mL 2YT. The culture was incubated at 37 °C with shaking until $OD_{600}$ ~ 0.5, infected with 1 × 10¹⁰ pfu/mL M13K07 helper phage and incubated for 30 min at 37 °C. The culture was centrifuged at 4800 × g for 10 min at 4 °C, the pellet resuspended in 500 μL 2YT and plated onto 24.5 cm square 2YT-Agar plates supplemented with 100 μg/mL carbenicillin and 50 μg/mL kanamycin. After overnight incubation at 30 °C, the bacterial-lawn obtained was resuspended into 25 mL TBS, and phage purified by precipitation with PEG/NaCl buffer. After two rounds of PEG/NaCl precipitation, the phage were then resuspended in PBS + 10% glycerol. From the second selection round onwards, the number of washes was increased to seven washes with BBB, seven washes with PBST, and two washes with PBS. The rest of the panning procedure was identical.

## Plating and sequencing of selected phage

To identify unique anti-EIF4e VH domains, the glycerol stocks for monoclonal screening were plated onto 2YT agar plates supplemented with 100 μg/mL Carbenicilin and incubated overnight at 37 °C. Individual colonies were infected with 1 × 10¹⁰ pfu/mL M13K07 helper phage and grown in 1 mL 2YT broth supplemented with 100 μg/mL Carbenicilin and 50 μg/mL Kanamycin overnight at 30˚C. The cells were pelleted by centrifugation at 3300 × g for 15 mins at 4 °C, and the supernatant used for phage monoclonal ELISA. To do so, NeutrAvidin was immobilized at 5 μg/mL onto a Maxisorp 96-well plate (Thermo Scientific) overnight at 4 °C, washed twice with PBS and blocked for 1 h at RT with BBB. Biotinylated eIF4E was added at a concentration of 2 μg/mL, and incubated for 1 h at RT. The plate was washed twice with PBS, and 25 μL of culture was mixed with 25 μL of block buffer, added to the plate and incubated for 2 h at RT. The plate was washed eight times with PBST, 50 μL of anti-M13 antibody HRP conjugate (Cytiva) was added at a 1:7000 dilution in block buffer, and incubated for 1 h at RT. The plate was washed eight times with PBST, and developed with 50 μL 3,3′,5,5′-Tetramethylbenzidine (TMB) substrate (GeneTex). After 5–15 min, the reactions were stopped by adding 50 μL of H₂SO₄, and signal was measured at an absorbance of 450 nm. Monoclonal cultures leading to high signal intensity (Absorbance higher than 1) were sequenced by Sanger sequencing to identify the unique VH domains binding to the target.

## Fluorescence competition assays

eIF4E:4G Binding Site: Competitive fluorescence experiments were carried out with the concentration of eIF4E constant at 200 nM and the labelled peptide (Ac-KKRYSRDFLLALQK-(FAM)-NH₂, Mimotopes) at 50 nM. Candidate VH domains were then titrated against the complex of the FAM labelled peptide and eIF4E.

m⁷GTP Binding Site: Competitive fluorescence experiments were carried out with the concentration of eIF4E constant at 200 nM and carboxyfluorescein (FAM) labelled m⁷GTP (JenaBiosciences) at 50 nM. Candidate VH domains were then titrated against the complex of the FAM labelled m⁷GTP and eIF4E. The $IC_{50}$ values were then determined for the candidate VH domains by fitting the experimental data to a four-parameter logistic regression model shown below in (1):

$$y = d + \frac{a - d}{1 + \left(\frac{x}{c}\right)^b} \tag{1}$$

$x$ = denotes the candidate VH domain concentration and $y$ = measured fluorescence anisotropy. The four parameters derived from the fitting procedure performed in Prism 8.0 (GraphPad Software): $a$ = the fluorescence anisotropy value at minimum dose, $d$ = maximum fluorescence anisotropy value measured at maximum dose, $c$ = the point of inflection (IC50) and $b$ = the Hill's coefficient. The tracer peptide and FAM labelled m⁷GTP were dissolved in DMSO at 1 mM and diluted into experimental buffer. Readings were carried out with a Envision Multi-label Reader (PerkinElmer). Experiments were carried out in PBS (2.7 mM KCl, 137 mM NaCl, 10 mM Na₂HPO₄ and 2 mM KH₂PO₄ (pH 7.4)) and 0.1% Tween 20 buffer. All titrations were carried out in triplicate.

## Screening eIF4E binding by biolayer interferometry (BLI)

Biotinylated eIF4E was immobilized on Streptavidin Biosensors (ForteBio/Satorius), and the VH clones were assessed for binding for 120 s at a single concentration of 2.5 μM in Kinetic Buffer (PBS + 0.02% Tween20, 0.1% BSA, 0.05% sodium azide), followed by a 120 s dissociation in the same buffer without VH. BLI measurement were made using a Blitz analysis system (ForteBio/ Satorius). The equilibrium constant ($K_d$) was calculated using Blitz Pro software (ForteBio/ Satorius).

## eIF4E immobilisation for surface plasmon resonance (SPR) binding assays

Pure eIF4E was immobilized on a CM5 sensor chip. The CM5 chip was conditioned with a 6 s injection of 100 mM HCL, followed by a 6 s injection of 0.1% SDS and completed with a 6 s injection of 50 mM NaOH at a flow rate of 100 μl/min. Activation of the sensor chip surface was performed with a mixture of NHS (115 mg ml⁻¹) and EDC (750 mg ml⁻¹) for 7 min at 10 μl min⁻¹. Purified eIF4E was diluted with 10 mM sodium acetate buffer (pH 5.0) to a final concentration of 0.5 μM with m⁷GTP present in a 2:1 ratio to stabilize eIF4E. The amount of eIF4E immobilized on the activated surface was controlled by altering the contact time of the protein solution and was approximately 250 RU. After the immobilization of the protein, a 7-min injection (at 10 μl min⁻¹) of 1 M ethanolamine (pH 8.5) was used to quench excess active succinimide ester groups. Six buffer blanks were first injected to equilibrate the instrument fully. Surface Plasmon resonance experiments were performed on a Biacore T100 machine. Stock protein solutions were serially diluted into running buffer immediately prior to analysis. Running buffer consisted of 10 mM Hepes pH 7.6, 0.15 M NaCl, 1 mM DTT and 0.1% Tween20.

## Surface plasmon resonance (SPR) multi-cycle injection experiments

Multi-cycle injection experiments were performed using a flow rate of 50 μl/min, compounds were injected for 60 s and dissociation was monitored for 180 s. Individual proteins were injected across the CM5 chip in threefold dilution series to at appropriated concentration to determine their respective binding constants. Each independent protein injection sampled one concentration only and was immediately followed by a similar injection of SPR buffer to enable the chip surface to be full regenerated by dissociation. Any protein which possessed extremely slow off-rates, and thus making dissociation an unsuitable method for regenerating the CM5 chip surface, were analysed using single-injection cycles. Responses from the target protein surface were transformed by: (i) subtracting the responses obtained from the reference surface that contained no immobilised protein, and (ii) subtracting the responses of the buffer injections from those of peptide injections. The last step is known as double referencing, which corrects the systematic artefacts. $K_d$s were determined using the BiaEvaluation software (Biacore) and calculated from both the response of the eIF4E coated CM5 chips at equilibrium and kinetically from the dissociation and association phase data for each of the peptides. Both the equilibrium and kinetic data were fitted to 1:1 binding model. Each individual peptide $K_d$ was determined from three separate titrations. Within each titration at least two concentration points were duplicated to ensure stability and robustness of the chip surface. Data analysis performed with Biacore T100 evaluation software (v2.0.4).

## Surface plasmon resonance (SPR) single injection experiments

Proteins that possessed extremely slow off-rates were analysed using a single-cycle kinetics approach. For each independent protein, samples are injected consecutively across the CM5 chip surface at different concentrations with no intervening regeneration steps. Concentrations used sampled a threefold dilution series with the series proceeding from the lowest concentration to the highest. A complementary single injection was performed over a reference surface containing no immobilised eIF4E. The response from the target protein surface was then transformed by: (1) subtracting the response the control surface and then (2) the response from of the buffer injections from those of the protein injections. The transformed response was analysed using Biacore T100 evaluation software (v2.0.4) to derive the binding and kinetic parameters using a 1:1 binding model.

## Expression and purification of GST-fused eIF4E and 4E-BP1⁴ᴬᴸᴬ

eIF4E and 4EBP1⁴ᴬᴸᴬ mutants were cloned into the GST fusion expression vector pGEX-6P1 (GE Lifesciences). BL21 DE3 competent bacteria were then transformed with the GST-tagged fusion constructs. A single colony was picked and transformed cells were grown in LB medium at 37 °C to an OD600 of -0.6 and induction was carried out overnight with 0.3 mM IPTG at 16 °C. Cells were harvested by centrifugation, and the cell pellets were resuspended in PBS (phosphate-buffered saline, 2.7 mM KCl and 137 mM NaCl, pH 7.4) and then sonicated. The sonicated sample was centrifuged for 60 min at 17,000 × g at 4 °C. The supernatant was applied to a 5 ml FF GST column (Amersham) pre-equilibrated in PBS buffer with 1 mM DTT. The column was then further washed by 6 volumes of PBS. Proteins were then purified from the column by cleavage with PreScission (GE Lifesciences) protease. Ten units of PreScission protease, in one column volume of PBS with 1 mM DTT buffer, were injected onto the column. The cleavage reaction was allowed to proceed overnight at 4 °C. The cleaved protein was then eluted off the column with wash buffer. Protein fractions were analysed with SDS page gel and concentrated using a Centricon (3.5 kDa MWCO) concentrator (Millipore). Protein samples were then dialyzed into a buffer solution containing 20 mM Tris pH 8.0 with 1 mM DTT and loaded onto a mono Q column pre-equilibrated in buffer A (20 mM Tris, pH 8.0, 1 mM DTT). The column was then washed in 6 column volumes of buffer A and bound protein was eluted with a linear gradient of 1 M NaCl over 25 column volumes. Protein fractions were analysed with SDS page gel and concentrated using a Centricon (3.5 kDa MWCO) concentrator (Millipore). The cleaved constructs were then purified to 90% purity. Protein concentration was determined using A280.

## Sortase (SrtA⁸ᴹ) expression and purification

The protein sequence corresponding to 61–206 of SrtA (Staphylococcus aureus) containing the following mutations (P94R, D160N, D165A, K190E, K196T, E105K, E108A and G167E) was ordered as a gene fragment from IDT (Integrated DNA technologies). The sequence was PCR amplified and inserted into a pNIC-CH bacterial expression plasmid via ligation-independent cloning in frame with a C-terminal 6xHis tag. The pNIC-CH-(61-206)SrtA⁸ᴹ (termed SrtA⁸ᴹ) expression vector was transformed into BL21(DE3) Rosetta competent cells and a single colony was used to inoculate a 20 ml starter culture in TB (terrific broth containing 25 μg/ml of chloramphenicol and 20 μg/ml of kanamycin), which was incubated overnight at 37 °C and shaken at 200 rpm. The starter culture was used to inoculate 750 ml of TB and was incubated at 37 °C until a O.D₆₀₀ reading of 2.0 was attained. Where upon the temperature of the culture was lowered to 18 °C and protein expression induced with 0.5 mM of IPTG overnight. Cells were harvested by centrifugation, and the cell pellets were re-suspended in 20 mls of lysis buffer (100 mM HEPES pH 8.0, 500 mM NaCl, 10 mM imidazole, 10% glycerol, 0.5 mM TCEP, 1000u Benzonase (Merck)) and then sonicated. The sonicated sample was centrifuged for 30 min at 17,000 × g at 4 °C. Supernatants were then filtered through 1.2 μm syringe filters and were loaded onto a Ni-nitrilotriacetic acid (NTA) column, pre-equilibrated with 20 mM HEPES pH 7.5, 100 mM NaCl, and 0.5 mM TCEP. The column was then washed with 5 column volumes of the same buffer containing 10 mM imidazole. Hexahistidine tagged SrtA⁸ᴹ was then eluted with a 1 M imidazole linear gradient. The protein was further purified by size exclusion chromatography (HiLoad 16/60 Superdex 75 prep grade, Cytiva Lifescience) using a 20 mM HEPES pH 7.5, 300 mM NaCl, 10% (v/v) glycerol, 0.5 mM TCEP, buffer. Protein concentration was determined using A280 with an extinction coefficient determined from the primary sequence of the construct determined by Prot-PARAM (https://web.expasy.org/protparam).

## N-terminal biotin labelling of eIF4E mediated by SrtA⁸ᴹ

Sortase-mediated ligation was used to specifically label eIF4E at the N-terminal with biotin. Cleavage of the GST-fused eIF4E with thrombin leaves a single glycine at the N-terminal. The ligation was carried out with thrombin cleaved eIF4E at 50 μM, SrtA⁸ᴹ at 1 μM, and biotin-KGGGLPET-GG-OHse(Ac)-amide peptide at 200 μM in 200 μL of ligation buffer (50 mM Tris pH 8.0, 150 mM NaCl, 1 mM TCEP). SortaseA61-206/8 M contains mutations that increase ligation efficiency[66] and make it calcium-independent[67]. The ligation was incubated at room temperature for 4 h. SrtA⁸ᴹ which contains a C-terminal 6×His-tag was removed with Dynabeads His-Tag (cat# 10104D, Thermo Fisher). The biotinylated protein was then dialyzed at 4 °C using slide-A-Lyzer cassette (10k MWCO) against 2 L of an appropriate buffer. The buffer was changed after 4−5 h and the dialysis was repeated for overnight. The biotinylated protein was aliquoted, snap-frozen with liquid nitrogen, and stored at −80 °C.

## Alanine scanning mutagenesis using yeast surface display (YSD)

Yeast display protocols were adapted from Wittrup and co-workers[68]. The VH-1C5 gene was ordered from Integrated DNA Technologies and the pCT-CON vector was digested using SalI, NheI, and BamHI restriction enzymes (NEB) to ensure complete linearization and absence of full-length insert therefore preventing transformation of yeast cells with parental plasmid. The VH-1C5 gene was then PCR

amplified using primers containing 50 base pairs of homology to the pCT-CON2 vector. 300 ng of VH-1C5 gene and 1 μg of plasmid vector were combined with 50–100 μL of electrocompetent EBY100 yeast cells and electroporated at 0.54 kV and 25 μF. Homologous recombination of the linearized vector and VH-1C5 insert yielded intact plasmid. Cells were grown in YPD (1% yeast extract, 2% peptone, 2% glucose) for 1 h at 30 °C, 250 rpm. The pCTCON2-VH-1C5 plasmid was then isolated and purified from EBY100 yeast cells using a Zymoprep kit II and then cleaned using the Qiagen PCR Purification kit. The CDR3 loop of the VH-1C5 gene was then sequentially mutated to alanine via site-directed mutagenesis using an in-fusion HD cloning Plus kit (Takara Bio), following the manufacturer's instructions, to generate the following plasmids pCTCON2-VH-M1 to VH-M20. This corresponded to an alanine scan of the following range of amino acids in the VH-1C5 scaffold: 100–120.

pCTCON2-VH1-C5 alanine scanning mutants were then electroporated into electrocompetent EBY100 yeast cells (0.54 kV and 25 μF). Individual transformed yeast clones were then grown in SD-CAA, pH 5.3 3 (0.07 M sodium citrate, pH 5.3, 6.7 g/L yeast nitrogen base, 5 g/L casamino acids, 20 g/L glucose, 0.1 g/L kanamycin, 100 kU/L penicillin, and 0.1 g/L streptomycin), at 30 °C, 250 rpm to logarithmic phase, pelleted, and resuspended to $1 \times 10^7$ cells/mL in SG-CAA, pH 6.0 (0.1 M sodium phosphate, pH 6.0, 6.7 g/L yeast nitrogen base, 5 g/L casamino acids, 19 g/L dextrose, 1 g/L glucose, 0.1 g/L kanamycin, 100 kU/L penicillin, and 0.1 g/L streptomycin) to induce protein expression. Induced cells were grown at 30 °C, 250 rpm for 12–24 h. Yeast were then pelleted, washed in 1 ml PBSA (0.01 M sodium phosphate, pH 7.4, 0.137 M sodium chloride, 1 g/L bovine serum albumin), resuspended in PBSA to a density of $1 \times 10^7$ cell per ml and then added to individual tubes corresponding to each VH-1C5 alanine scanning mutant.

Purified sortase biotinylated eIF4E was then added to each mutant at a concentration of 2 μM and samples were incubated at 20 °C for 1 h. Cells were then pelleted by centrifugation ($14,000 \times g$ for 30 s at 4 °C), the supernatant aspirated and then washed with 1 ml ice-cold PBSA. Yeast were resuspended in 500 μl PBSA containing Anti-HA Ab Alexa Fluor 488 (ThermoFisher Scientific, catalogue #: A-11039, dilution used 1:200) and Streptavidin-phycoerythrin or neutravidin-phycoerythrin (ThermoFisher Scientific, Catalogue #: S866 and A26660, respectively) and incubated for 30 min. Working dilutions used for Streptavidin-phycoerythrin and neutravidin-phycoerythrin were 1:100 and 1:50, respectively. Cells were then pelleted at $14,000 \times g$ for 30 s at 4 °C, aspirate supernatant and wash with 1 ml PBSA buffer. Each VH-1C5 alanine mutant was then analysed by flow cytometry using an Aria (Becton Dickinson) cytometer and the BD FACSDiva6.1 software package. Cells positive anti-HA and eIF4E were selected and mean fluorescence intensity determined. FACs gating strategy shown in Fig. S15.

### Yeast surface display (YSD) affinity maturation

The library for affinity maturation was prepared using pCT-CON2-VH-M4 as a template for error-prone PCR. Error prone PCR was performed using conditions to introduce on average 1 or more amino acid change per 500 base pairs[69]. PCR primers containing 50 base pairs of homology to the pCT-CON2 vector. Multiple aliquots of ~10 μg of mutagenized pCT-CON2-VH1C5[M4] and 3 μg of linearised pCT-CON2 plasmid vector were combined with 50–100 μL of electrocompetent EBY100 and electroporated at 0.54 kV and 25 μF. Homologous recombination of the linearized vector and degenerate insert yielded intact plasmid. Cells were grown in YPD (1% yeast extract, 2% peptone, 2% glucose) for 1 h at 30 °C, 250 rpm. The number of total transformants was $5.7 \times 10^7$ cells as determined by serial dilutions plated on SD-CAA plates (0.1 M sodium phosphate, pH 6.0, 182 g/L sorbitol, 6.7 g/L yeast nitrogen base, 5 g/L casamino acids, 20 g/L glucose). The library was propagated by selective growth in SD-CAA, pH 5.3 (0.07 M sodium citrate, pH 5.3, 6.7 g/L yeast nitrogen base, 5 g/L casamino acids, 20 g/L glucose, 0.1 g/L kanamycin, 100 kU/L penicillin, and 0.1 g/L streptomycin) at 30 °C,

250 rpm. Where upon it was pelleted, and resuspended to $1 \times 10^7$ cells/mL in SG-CAA, pH 6.0 (0.1 M sodium phosphate, pH 6.0, 6.7 g/L yeast nitrogen base, 5 g/L casamino acids, 19 g/L dextrose, 1 g/L glucose, 0.1 g/L kanamycin, 100 kU/L penicillin, and 0.1 g/L streptomycin) to induce protein expression. Induced cells were grown at 30 °C, 250 rpm for 12–24 h. Yeast were then pelleted, washed in 1 mL PBSA (0.01 M sodium phosphate, pH 7.4, 0.137 M sodium chloride, 1 g/L bovine serum albumin), resuspended in PBSA to a density of $1 \times 10^7$ cell per ml and then used for affinity maturation described in the paragraph below:

FACS selections of the mutagenized VH-M4 library was conducted with the kinetic competition. Three rounds of kinetic selection were performed. Yeast were washed and incubated with 200 nM of biotinylated eIF4E. Yeast were then washed and resuspended with PBSA (phosphate-buffered saline with bovine serum albumin) 2 μM of unbiotinylated eIF4E (to prevent further association of labelled target) and incubated at room temperature for 8 minutes to enable dissociation of biotinylated eIF4E. Cells were washed in PBSA, resuspended in PBSA with Anti-HA Ab Alexa Fluor 488 (ThermoFisher Scientific, catalogue #: A-11039, dilution used 1:200)and Streptavidin-phycoerythrin or neutravidin-phycoerythrin (ThermoFisher Scientific, Catalogue #: S866 and A26660, respectively)for 10 min, and incubated on ice. Working dilutions used for Streptavidin-phycoerythrin and neutravidin-phycoerythrin were 1:100 and 1:50, respectively. Labelled cells were washed with 1 mL PBSA, resuspended in 0.5–2.0 mL PBSA and analysed by flow cytometry using an Aria (Becton Dickinson) cytometer in conjunction with the BD FACSDiva6.1 software package. Cells positive for anti-HA and eIF4E were selected and sorted. FACs gating strategy shown in Fig. S15. Collected cells were grown in SD-CAA, pH 5.3, at 30 °C, 250 rpm and either induced in SG-CAA, pH 6.0, for further selection or used for plasmid recovery. Two further rounds of kinetic selection were performed as described above extending the periods of dissociation to 60 and 120 min, respectively. Yeast isolated from each round for plasmid recovery were serially diluted and plated on SD-CAA plates (0.1 M sodium phosphate, pH 6.0, 182 g/L sorbitol, 6.7 g/L yeast nitrogen base, 5 g/L casamino acids, 20 g/L glucose) and individual clones grown. Plasmid DNA was then isolated using the Zymoprep kit II, cleaned using the Qiagen PCR Purification kit, and transformed into DH5α (Invitrogen) cells. Purified plasmids were then sequenced using BigDye chemistry. Fifty clones were sequenced from each affinity maturation round of selection. The final round of kinetic selection resulted in the identification of only three eIF4E binders: VH-1C5[D104A/S108R/F120I], VH-1C5[D104A/S24G/F120I] and VH-1C5[D104A/Y97C/F120I]. The VH-1C5[D104A/S108R/F120I] clone (termed VH-S4) dominated the final round with a frequency > 90% and was therefore selected for scale up and SPR analysis. The clone was then amplified using PCR and ligated into the pET22B bacterial protein expression vector for protein purification as outlined in "Bacterial Expression and Purification of VH Domains" section.

### Bacterial expression and purification of VH domain binders

VH-1C5 and VH-1A2 sequences were ordered as gene fragments from Integrated DNA Technologies (IDT). Both coding sequences were PCR amplified and cloned directly into the bacterial expression vector pET-22b(+) with an in-frame C-terminal six-hisitidine tag. VH-M4 was directly PCR amplified from the pETCON2 plasmid used in the yeast alanine scanning experiments, whilst VH-S4 was amplified from the plasmid isolated through the affinity maturation selection. Both sequences were then cloned into pET-22b(+) as described earlier. Using VH-M4 as a template sequence, the in-fusion mutagenesis kit (Takara) was used to generate the following mutants in the pET-22b(+) backbone (VH-1C5[D104A/S108R] and VH-1C5[D104A/F120I]). Each VH domain plasmid was separately transformed into *E. coli* BL21 (DE3) cells and used to inoculate 10 mls of LB broth (containing 100 μg/ml) started culture, which were incubated overnight before being used to seed

**Table 3 | Primer sequences used for construction of expression plasmids**

| Primer name | Primer sequence |
|---|---|
| pCDNA3 VH Fw NheI | CCCAAGCTGGCTAGCATGAGCGAAGTGCAGCTGGT |
| pCDNA3 VH Rv BamHI | TGGACTAGTGGATCcATTCCGCCGCTGCTCACGGT |
| pCW57 VH Fw AvrII | AACCCCGGTCCTAGGatgagcgaagtgcagctg |
| pCW57 VH Fw BamHI | CCCAACCCCGGATCCTCACTTGTCATCGTCATCCTTGTAATC |
| peT22B VH Fw BamHI | CGGAATTAATTCGGATCCATGGAAGTGCAGCTGGT |
| peT22B VH Fw XhoI | GTGGTGGTGGTGCTCGAGCCGCCGCTGCTCACGGT |
| pCDNA3 4EBP1 Fw NheI | CCCAAGCTGGCTAGCgccATGAGTGGCGGATCATCCT |
| pCDNA3 4EBP1 Rv BamHI | CTGGACTAGTGGATCAATATCCATTTCGAACTGGGACTC |
| pCW57 4EBP1 Fw AvrII | AACCCCGGTCCTAGGATGAGTGGCGGATCATCCT |
| pCDNA3 VH Fw NheI | CCCAAGCTGGCTAGCATGAGCGAAGTGCAGCTGGT |
| pCDNA3 VH Rv BamHI | TGGACTAGTGGATCcATTCCGCCGCTGCTCACGGT |

1000 mls of fresh LB broth. Bacterial cultures were grown at 37 °C and when they reached a $OD_{600}$ of 0.6–0.8, the cells were induced with a final concentration 0.5 mM of IPTG and incubated overnight at 25 °C. Cells were harvested by centrifugation at $17,000 \times g$ for 10 min and pellets were resuspended in lysis buffer (25 mM HEPES pH 7.5, 300 mM NaCl, 20 mM imidazole, 1 mM DTT) and then sonicated for 5 min. Bacterial supernatants were then filtered through 1.2 μm syringe filters. Proteins were purified through a standard two-steps protocol: first, supernatant were loaded onto a 1 ml HisTrap column (Cytiva Lifesciences), which was pre-equilibrated then extensively washed with buffer A (25 mM HEPES pH 7.5, 300 mM NaCl, 1 mM DTT) and then eluted with buffer A that also contained 500 mM imidazole; second, the eluted proteins were subjected to gel filtration chromatography on a Superdex 75 column (Cytiva Lifesciences) using PBS buffer containing 1 mM DTT. Protein fractions were analysed by SDS page gel and concentrated. Protein concentration was determined using absorbance at A280 nm.

## Protein crystallization
The eIF4E:VH-1C5 and eIF4E:VH:VH-S4 complexes were crystallized by vapour diffusion using the sitting drop method. Crystallization drops contained eIF4E, $m^7GTP$ and VH-1C5 at concentrations of 100, 300 and 100 μM respectively. Sitting drops were set up in 48 well Intelli-Plates (Hampton research) with 1 μl of the protein sample mixed with 1 μl of the mother-well solution. eIF4E:VH-1C5 crystals grew over a period of one week in 25% PEG6000, 0.01-02 M Ammonium Sulphate and 100 mM Tris at pH 8.5. eIF4E:VH-S4 crystals grew over a similar period of time but in 0.01 M Tri-sodium citrate and 16% PEG6000 (v/v). Crystals containing the $m^7GTP$:eIF4E:VH-S4ss complex were isolated in 0.1 M Sodium HEPES pH7.5, 25% PEG 6000. For X-ray data collection at 100 K, crystals for all sets of crystallization conditions were transferred to an equivalent mother liquor solution containing 25% (v/v) glycerol and then flash frozen in liquid nitrogen.

## Crystal data collection and refinement
X-ray diffraction data were collected at the Australian synchrotron (MX1 beamline) using a CCD detector, and integrated and scaled using XDS. The initial phases of the VH-domain complexed crystals of eIF4E were solved by molecular replacement with the program PHASER[70]

using the human eIF4E structure (PDB accession code: 1EJ4) and the VH domain structure (PDB accession code: 5TDP, chain B) as independent search models. The starting models were subjected to rigid body refinement and followed by iterative cycles of manual model building in Coot and restrained refinement in Refmac 6.0[71]. Models were validated using PROCHECK[72] and the MOLPROBITY webserver[73]. Final models were analysed using PYMOL (Schrödinger). See Table S2 for data collection and refinement statistics. The eIF4E complex structures with VH-1C5, VH-S4 and VH-S4ss has been deposited in the PDB under the submission codes 7D6Y, 7D8B and 7XTP, respectively. For data collection and refinement parameters see Table S3.

## Plasmid and reagents for cell biology
All plasmids were purchased from Addgene where not indicated otherwise. 4D5 VH domain plasmid was provided by DotBio. Mutant VH domains were generated with In-fusion mutagenesis kit (Clontech). VH domain (VH-1C5, VH-1A2, VH-M4, VH-S2, VH-S4, VH-S4ss and VH-1C5$^{Scr}$) nucleotide sequences are listed in Table S5. VH mutants were cloned into a pCDNA3.1 vector (Thermo Fisher Scientific) harbouring a C-terminal 3× FLAG tag via NheI/BamHI sites to allow mammalian cell overexpression. For bacterial expression, VH domain mutants were cloned into either a pET22B or pCW57 plasmid using either BamHI/XhoI or BamHI/AvrII cloning sites, respectively. pcDNA3-rLuc-polIRES-fLuc (bicistronic reporter), eIF4E and eIF4G$^{604-646}$ NanoBIT and 4E-BP1 mutant plasmids generation have been described previously[41]. PP242 and Staurosporine were purchased from Tocris Bioscience, whilst all other chemicals unless otherwise stated were purchased from Selleck Chemicals. Primer sequences used for construction of expression plasmids shown in Table 3.

## Mammalian cell culturing conditions
All cell lines were cultured in DMEM cell media supplemented with 10% foetal calf serum (FBS) and penicillin/streptomycin. For A375 and MDA-MB 231 stable cell lines medium FBS was replaced with TET-system approved FBS (Thermo Fisher Scientific). Cells were maintained in a 37 °C humidified incubator with 5% $CO_2$ atmosphere. A375, HELA and MDA-MD-231 cell lines were purchased from ATCC (catalogue numbers: CRL-1619, CCL-2 and HTB-26, respectively). The 293FT cell lines were sourced from Thermo Fisher Scientific (catalogue number: R70007). All cells were routinely tested for mycoplasma infection every 3 weeks.

## Generation of VH-S4 and 4E-BP1$^{4ALA}$ inducible stable cell lines
Confluent HEK293FT cells were used to generate lentivirus for infection of target cells. Packaging cells were transfected using calcium phosphate transfection described by Trono laboratory (https://www.epfl.ch/labs/tronolab). 6 μg of pCW57 plasmid (Addgene, USA) harbouring either 4EBP1$^{4ALA}$ or VH-S4 or no insert were co-transfected into HEK293T cells with plasmids encoding pLVSVG (viral envelope), pLP1 (gag-pol) and pLP2 (rev), in a ratio of 2:1:2:2 to generate viral particles. 48 h later the conditioned medium harbouring viral particles from the transfected HEK293T cells was filtered and viral particles were concentrated by ultracentrifugation. A375 and MBA-MD-231 cells were seeded in 12-well plates and infected with viral particles over a 12 h period prior to cell media replacement with fresh medium. 72 h post infection, A375 and MBA-MD-231 cells were supplemented with 800 μg/ml of geneticin and selections for stably transfected cells were carried out for 2 weeks, replacing the antibiotic-containing media every 3 days. Polyclonal geneticin-resistant pools of cells were then obtained. These were then incubated with 1 μg/ml of doxycycline for 24 h, where upon GFP positive single clones were isolated by FACs into 96-well plates. Monoclonal stable cell lines were verified using western blot and then expanded for subsequent analysis.

**Table 4 | List of antibodies used in cell biology studies with experimental dilutions**

| Antibody | Catalogue number | Clone | Company | Dilution used |
|---|---|---|---|---|
| Anti-eIF4E (phospho S209) | AB76256 | EP2151Y | Abcam | 1:5000 |
| Anti phospho 4EBP1 threonine 36/46 | 2855 | 236B4 | Cell signalling Technology | 1:1000 |
| Anti p44/42 MAPK | 9102 | Not provided | Cell signalling Technology | 1:1000 |
| Anti pAKT ser473 | 9271 | Not provided | Cell signalling Technology | 1:1000 |
| Anti eIF4E | 2067 | C46H6) | Cell signalling Technology | 1:500 |
| Anti 4EBP1 | 9644 | 53H11 | Cell signalling Technology | 1:4000 |
| Anti phospho p44/42 |  | Not provided | Cell signalling Technology | 1:1000 |
| Anti AKT |  | Not provided | Cell signalling Technology | 1:1000 |
| Anti eiF4G1 | 2858 | Not provided | Cell signalling Technology | 1:500 |
| Anti cyclin D1 | 2988 | 92G2 | Cell signalling Technology | 1:500 |
| Anti Mcl-1 | 5453 | D35A5 | Cell signalling Technology | 1:1000 |
| Anti 4EBP2 | 2845 | Not provided | Cell signalling Technology | 1:1000 |
| Anti Bcl-Xl | 2764 | 54H6 | Cell signalling Technology | 1:1000 |
| Anti Parp | 9532 | 46D11 | Cell signalling Technology | 1:1000 |
| Anti Stat-1 | 9172 | Not provided | Cell signalling Technology | 1:1000 |
| Anti β-actin peroxidase | A3854 | AC-15 | Sigma aldrich | 1:10,000 |
| Anti-FLAG peroxidase | A8592 | M2 | Sigma aldrich | 1: 1000 |
| Anti Phospho-S6 Ribosomal subunit Ser235/236 | 2211 | Not provided | Cell signalling Technology | 1: 1000 |
| Anti S6 Ribosomal subunit | 2317 | 54D2 | Cell signalling Technology | 1: 1000 |

### Immunoprecipitation and m7GTP pull down experiments

Twenty-four hours prior to transfection or drugging, cells were seeded at a cell density of 1,000,000 (HEK293) or 250,000 (Hela) or 300,000 (A375) cells per well of a six-well plate (ThermoFisher Scientific). Transfections were performed using Lipofectamine 3000 (Thermo-Fisher Scientific) with either 1 µg or the indicated amount of plasmid vectors per well according to the manufacturer's instructions. After a 48 h (or as indicated in the relevant figure) incubation period, the cell media was then removed and the cells washed with PBS saline. Cells were directly lysed in the wells with 300 µl of lysis buffer containing 20 mM Hepes pH 7.4, 100 mM NaCl, 5 mM $MgCl_2$, 0.5% NP-40, 1 mM DTTl with protease (Roche) and phosphatase (Sigma-Aldrich) inhibitor cocktail sets added as outlined by the manufacturer's protocols. Cellular debris was removed by centrifugation, and the protein concentration was then determined using the BCA system (Pierce). m7GTP pulldown and FLAG immunoprecipitation experiments were performed with 200 µg of cell lysate, which was either incubated with 20 µl of m7GTP (Jena Bioscience) or anti-FLAG M2 antibody (Roche) immobilised agarose beads for 2–4 h at 4 °C on a rotator. Beads were then washed four times with lysis buffer containing no protease or phosphatase inhibitors. This was then followed by the addition of Laemlee buffer (2×) and the beads boiled for 5 min at 95 °C. Samples were centrifuged and the supernatant removed for western blot analysis.

### NanoBit® eIF4E:eIF4G604–646 complementation assay

Opaque 96-well plates were seeded with 30,000 HEK293 cells per well in DMEM and 10% FCS. Transfections in 96-well plate format were performed using FUGENE6 (Roche) with 30 ng total DNA of the two NanoBit plasmid vectors (eIF4G-LgBiT and SmBiT-eIF4E in a 1:1 ratio) and 100 µg of the indicated plasmid per well. 48 h after transfection, the medium was replaced with 100 µl of Opti-MEM cell media containing 0% FCS with no added red phenol (Thermo Fisher Scientific) and luminescence activity was assayed as described elsewhere[41] by an Envision Multi-Plate reader.

### Cap-dependent translation and AlphaScreen®Surefire® assays

Opaque 96-well plates were seeded with 30,000 HEK293 cells per well in DMEM and 10% FCS. Transfections were performed using FUGENE6

(Roche) with 30 ng of the bicistronic reporter (pcDNA3-rLuc-polIRES-fLuc) plasmid and 150 ng of the indicated plasmid. Forty-eight hours after transfection, Renilla and firefly luminescence activity was determined using the Dual Glo Luciferase Assay System (PRO-MEGA). A replicate plate following the experimental conditions above was also concurrently prepared, where the cells were instead lysed with 50 µl of passive lysis buffer (PerkinElmer) for 15 min and 10 µl of lysate from each well was transferred into a white bottom 384-well plate. GAPDH and pS209 eIF4E levels were then determined using the Alphascreen®Surefire® GAPDH and eIF4E (p-Ser209) assays (PerkinElmer) as outlined in the manufacturer's instructions. Luminescence readings were performed using an Envision Multi-plate reader (PerkinElmer).

### Mammalian cell protein expression analysis

Transfected HEK293 cells (prepared as described in the NanoBit and Cap-dependent translation Experiments sections) were seeded with 30,000 cells per well in 96-well plates. After an incubation period indicated in the relevant figure, cells were washed with PBS and directly lysed in the wells of the plate with 50 µl of cell lysis buffer (20 mM Hepes pH 7.4, 100 mM NaCl, 5 mM $MgCl_2$, 0.5% NP-40, 1 mM dithiothreitol) with protease (Roche) and phosphatase (Sigma-Aldrich) inhibitor cocktail sets added as outlined by the manufacturer's protocols. Cellular debris was separated by centrifugation, and without further quantification, samples were analysed by western blot.

### Western Blot analysis

Samples were resolved on midi or mini Tris-Glycine 4–20% gradient gels (Bio-Rad) according to the manufacturer's protocol. Western transfer was performed with an Immuno-blot PVDF or nitrocellulose membrane (Bio-Rad) using a Trans-Blot Turbo system (Bio-Rad). Western blots were then performed. B-actin levels were measured to ensure equal loading. For details of antibodies and experimental dilutions used see Table 4.

### Cell proliferation assay

A375 cell lines were plated in 96-well clear bottom plates at a density of 4000 cells per well in 200 ul DMEM and 10% FCS medium. After 24 h,

cell media was replaced with 200 μl of medium containing doxycycline at 1 ng/ml. Cell confluence and cell growth was then measured continuously over 7 days using an IncuCyte FLR instrument (EssenBioscience).

## Cell viability assay
Opaque 96-well plates were seeded with A375 cells and cells treated with doxycycline as described in Cell Proliferation Assay section. At the indicated time points, CellTiter-GLO 2.0 reagent (Promega) wasto each well and cellular viability determined according to the manufacturer's instruction. Luminescence readings were performed using an Envision Multi-plate reader (PerkinElmer).

## Global protein synthesis measurements
A375 cell lines were seeded in 12 well-plates at a cell density of 120,000 cells per well in 1 ml of DMEM and 10% FCS medium. The SUnSET assay was used to monitor the rate of protein synthesis (REF). Transfected cells were pulse labelled with the addition of 10 μg/ml of puromycin (ThermoFisher Scientific) to the cell media prior to cell lysis. As a control, cycloheximide (Sigma) was added at 10 μg/ml five minutes before puromycin addition, resulting in complete blockade of protein synthesis. All other drugs were added for the period indicated in the figures. Cell extracts were processed and analysed for western blotting using anti-puromycin antibody, as outlined in the western blot analysis section.

## Mass spectrometry analysis of intra-domain disulfide formation
Mass spectra were acquired in positive mode using a Shimadzu LCMS-2020 single quadrupole liquid chromatography-mass spectrometry system, equipped with a Phenomenex Aeris Widepore XB-C8 analytical column (2.1 × 50 mm; 3.6 μm packing). Mobile phase A was $H_2O$ (0.1% formic acid); mobile phase B was acetonitrile (0.1% formic acid). Chromatographic conditions were as follows:

Flow rate: 0.4 mL/min
Column oven: 25 °C
Gradient program (linear):

| Time (min) | Mobile phase B (%) |
| --- | --- |
| 0 | 5 |
| 3 | 5 |
| 18 | 65 |
| 21 | 65 |
| 24 | 5 |

Recombinant protein stocks (~5 mg/mL in phosphate-buffered saline) were diluted 10-fold into either: 6 M guanidine hydrochloride, 100 mM phosphate, pH 7 buffer ('denaturing, non-reducing' conditions); 6 M guanidine hydrochloride, 100 mM phosphate, pH 7 buffer containing 100 mM DTT ('denaturing, 100 mM DTT' conditions), or 1× phosphate-buffered saline ('native, 100 mM DTT' conditions). The resulting solutions were allowed to stand (ambient temp, 20 min), and then analysed by LC-MS (10 μL; ~5 μg). Mass spectra were integrated across the entire peak in the resulting total ion current chromatograms.

## Statistics and reproducibility
No statistical methods were used to predetermine sample size. The experiments were not randomised and investigators were not blinded to allocation during experiments and outcome assessment.

## Reporting summary
Further information on research design is available in the Nature Research Reporting Summary linked to this article.

## Data availability
Structure factors and coordinates generated in this study have been deposited in the Protein Data Bank under accession codes 7D8B, 7D6Y and 7XTP for the eIF4E:VH-1C5, eIF4E VH-S4 and eIF4E:VHS4ss complexes, respectively. The source data generated in this study are provided in the Supplementary information or Source Data file. Antibodies use in this study are provided in the supplementary information. Amino acid and nucleic acid sequences of the VH domains used in this study are listed in the supplementary information. Source data are provided with this paper.

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

## Acknowledgements

P.N. gratefully acknowledge funding from a start-up grant from Nanyang Technological University and grants from the Swedish Research Council. C.J.B., D.P.L. and C.S.V. are supported by the Agency for Science, Technology and Research (A*STAR).

## Author contributions

C.J.B., I.A, P.N., T.C., C.S.V., and D.P.L. conceived and designed the study. Y.F., M.G., and S.R.R. performed all cell biology experiments associated with the manuscript. Y.C.L carried out all the yeast surface display experiments and analysis of the affinity maturation results. K.H., A.H.E., A.P. and Y.X.C. performed the directed evolution experiments, constructed the phage library, carried out the phage panning and preliminary binding analysis of the initial selected clones. K.Y. undertook recombinant protein purification and thermal stability experiments. J.S. collected and processed X-ray crystallography as well as carrying out the SPR experiments and associated analysis. D.L. performed functional analysis of the elucidated crystal structures. Z.G. designed and undertook the mass spectrometry experiments as well as spectra analysis. D.T. performed recombinant protein purification and fluorescence polarization experiments. C.J.B. supervised the structural, biophysical and cellular studies. I.A. supervised directed evolution and phage selection experiments. C.J.B. and I.A. wrote the manuscript with input from all co-authors.

## Competing interests

P.N., I.A., K.H. and T.C. are shareholders of DotBio Pte. Ltd. I.A. K.H., A.H.E., A.P., K.Y and Y.X.C., are employees of DotBio Pte. Ltd. The stabilizing mutations introduced to the VH domains described in this paper, as well as the phage display libraries created, are the subject of the following patent applications: WO2016072938A1 (World Intellectual Property Organization, Patent Cooperation Treaty), US20170320934A1 (US Patent and Trademark Office, granted), US11053302B2 (US Patent and Trademark Office, granted), JP2018500879A (Japan Patent Office, granted), EP3215537A4 (European Patent Office, granted), EP3215537B1(European Patent Office, granted), EP4008729A1 (European Patent Office, pending) and CN107001477A (China National Intellectual Property Administration, pending). T.C., I.A. and P.N. are inventors in these patent applications, while Nanyang Technological University is the assignee. These patents are licensed exclusively to DotBio Pte. Ltd. Y.F., Y.C.L., J.S., S.R.R., M.G., D.L., Z.G., C.S.V., D.P.L., D.T. and C.J.B. have no competing interests to declare.
