## [Peer Review File · Nature Communications]

Reviewers' Comments:

Reviewer #1:

Remarks to the Author:

Asial, Brown and coworkers describe the construction of a VH only domain that is derived from therapeutic antibody trastuzumab that was engineering in several ways. First, a scaffold protein was generated with improved stability, from which the internal disulphide bond was removed, aimed at facilitating the use of these VH only proteins for in vivo applications. Then, by several steps of library screening and systematic analysis very potent binders were isolated that block eIF4E at the eIF4G interaction. The authors provide co-crystal structures of the VH inhibitor with the target and also performed sophisticated studies to proof the in vivo blocking function of the newly generated inhibitor. These are very interesting findings and the paper contains an impressive wealth of data that would merit publication. However, the manuscript lacks in many aspects clarity and requires extensive re-writing. Moreover, the title is too broad and should be more specific indicating that an inhibitor of eIF4E was isolated. Additionally, the introduction section is very short and very broad and the biological role of eIF4E in general and in malignancy and also the general interplay of eIF4E, eIF4F, eIF4G, eIF4BP1 etc. is not provided. This lack of information makes it very difficult to follow the logic of the in vivo experiments. Details should be given in the introduction section and a scheme/overview would be helpful to be provided at least in the supplemental. Another critical point is that the authors claim a new antibody scaffold for in vivo use and insinuate that absence of disulphide bonds is required for folding and intracellular function. This is obviously not (always) the case since numerous nanobodies exist, termed intrabodies that do contain a disulphide bond, are stable in vivo (see e.g. <https://pubmed.ncbi.nlm.nih.gov/33371447/>) and can also be considered for therapeutic applications in case that they can be delivered into cells. This should be considered and appropriately cited. I also wonder, whether reintroduction of the disulphide bond in VH-S4 results in an equally functional and possibly even more stable variant.

Points related to lack of clarity:

I recommend to mutually exchange Figures 1A and B and briefly mention in the text that the Hot-CoFi method is described in Figure 1A.

"Three residues in VH-36, C22 and 99 C92, as well as the core residue A24, were randomised with all 20 amino acids to generate a library 100 suitable for screening". Please explain why A24 was selected.

Li 127: "whilst CDR3 was randomized with sequences of different lengths and biased towards residues serine and tyrosine with approximately 20% frequency each." Please specify length distribution and how randomization was performed and also how the bias towards Ser and Tyr was achieved. How many individual clones were generated upon randomization of CDR1-3?

Phage Display library screening against eIF4E: How many different clones were obtained after how many screening rounds?

All Biacore sensorgrams: The concentrations range for measurements or the concentration for curve should be provided.

The function of 4EB1 should be explained upon first mentioning (see above). Likewise: The derived peptide is not cited.

Affinity maturation of VH-M4 binder by yeast display is not appropriately described. How many clones were generated? How many were screened? Details should be provided in the supplemental.

Sloppy wording, formatting and typos:

In Fig. 1A Hot CoFi screening is named as Hit CoFi Screening. Moreover, Figure 1 has both in the doc and in the PDF file a very bad resolution.

Li 95: Among these, VH-36 and VH-36 ... replace by VH-33 and VH-36.

Li 121: template, OR Ile, Val, Tyr, or Trp
Li 124, 190: Mixed upper and lower case.
Li 126: A library of VH domains WAS then
Li 130, 134, 165, 225, 227: figure should read Figure.
Legend to Figure 1: Kabaat should read Kabat.
Legend to Figure 2: shown on the graphs should read shown in the graph.
Inconsistent Figure labelling format: In Figure 2 A, B C is bold face, in Figure 3 not.
Legend to Figure 3: correct "methods)..D) Surface". CR3 loop should read CDR3 loop.
Legend to Figure 3D: VH-1C5E104D seems to be D104A !
Legend to figure 6: B-actin should read b-(font symbol) actin.
Li 297: Both miniproteins, caused dephos... remove comma.

Reviewer #2:

Remarks to the Author:

In this manuscript entitled "Engineering Disulphide-Free Autonomous Antibody VH Domains to modulate intracellular pathways", Frosi et al. utilize the 4D5 VH domain based on trastuzumab and through a series of well-designed selection assays engineer this domain into a mini-protein molecule that is disulphide-free, targets eIF4E with picomolar affinity, and is able to inhibit eIF4F translation in cells. They also define the molecular basis of the interactions between the mini-protein and eIF4E at the structural level by co-crystallization of the complexes. This is a really nice piece of work and I was impressed by the thoroughness of the work. The conclusions are well supported by the data.

I have a few important questions for the authors that I am sure could be addressed by relatively simple experiments.

1. How selective is the VH-1C5(D104A/S108R/F120I) evolved clone for eIF4E? For example, if the authors radiolabel cells expressing VH-1C5(D104A/S108R/F120I) with 35S-Met/Cys, and then pull down VH-1C5(D104A/S108R/F120I) they should see eIF4E following SDS-PAGE and autoradiography. Are there other proteins that are pulled down? One might expect to see eIF4A, eIF4G, and/or 4E-BPs depending on the stringency of the buffer used, but other than this – no other proteins should be pulled down. This experiment would provide a sense of what other "targets" might be engaged by VH-1C5(D104A/S108R/F120I). Whatever the answer, it is important to know.

2. Do the authors have a sense of the stoichiometry (at the protein level) of VH-1C5(D104A/S108R/F120I) required to dissociate eIF4E from eIF4G. Also, what is the subcellular localization of VH-1C5(D104A/S108R/F120I)? Any VH-1C5(D104A/S108R/F120I) in the nucleus where eIF4E has been implicated in mRNA transport?

3. Does recombinant VH-1C5(D104A/S108R/F120I) or other variant function in vitro when recombinant protein is added to translation extracts. What is the impact in vitro on cap-dependent versus cap-independent translation in vitro (I know the authors have done the experiments in cells, I am wondering if there is selectivity in vitro).

4. In Fig 5C, please present the FLuc and RLuc values as separate bar graphs. Information is lost when these are presented as ratios. The title of the panel should not be there since there are also results on the IRES in this graph.

I think the authors can make the MS a bit more accessible to a broader readership. My suggestion is that the authors revisit the the writing of the paper (at least in some sections). I believe that many labs would be interested in the overall pipeline described by the authors. One area that could use improvement is that they use several specialized technology (and corresponding terms) which are not described in their text. Examples are CoFi, Hot-CoFi, DSF, NanoBit [sometimes also spelt NanoBIT in the text], Surefire assay,.... A few introductory sentences as to what these approaches achieve and measure would be helpful here to a reader less knowledgeable with these approaches.

I ask the that nucleotide sequences of the different engineered mini-proteins (especially VH-S4) be included in an Appendix. Without these it becomes difficult for labs to reproduce and build on this

nice piece of work. The amino acid sequence is a start, but codon usage or bias cannot be assessed from solely amino acid sequence.

The MS should be carefully proof-read. There are many mistakes, some of which I list below
p.4. "The VH-36i.1 clone and its parent templates (VH-36 and VH-36i) all contained the mutation A100bP within the CDR-H3 loop. " What is a A100bP mutation?"

p. 5 " ... P100b was removed and replaced by the sequence SSSA" What is P100b?

Legend to Fig 5. "eiF4E" should read "eIF4E"

p. 12 "...ERK and rS6 phosphorylation compared". Do the authors mean rpS6?

Sometimes I read "4E-BP14AIA ", other times " 4E-BP14ALA

Proof the Supplemental information as well. For example in the legend to Figure S6, I found the following mistakes. "anti-bodies " "ant-FLAG." "B-Actin " "cyclohexamide " "X ug of whole cell lysates were analysed for puromycin labelling wing ant-Puoro."

Reviewer #3:

Remarks to the Author:

This manuscript describes the development of a cysteine-free VH framework, the construction of a phage-display library using the framework, the development of a series of high-affinity binders to eIF4E, detailed structural characterization of the binding interface, and the demonstration of the utility of one binder to modulate cellular functions of eIF4E.

This manuscript includes several strengths.

1. The development of a cysteine-free VH framework that has good "developability" may be of general interest to the protein-engineering field.
2. The phage-display library may be useful for generating binders to diverse targets.
3. The high-affinity binders to eIF4E with associated structural data may be of interest to those who study this specific system.

However, there are a number of weaknesses that need to be addressed.

1. This system joins a sizable group of already established molecular scaffolds that can be used for intracellular application, including nanobody, monobody and DARPIn to name just a few most advanced systems (PMID 20010839, 33371447, 32145686 and 32591521 and references therein). As such the novelty of the system itself is limited. The statement, "Many currently used miniproteins are limited as binding modules due to randomization of highly rigid structural motifs with limited sites of variation." (page 2) seems to contradict with the large body of the literature. The manuscript does not cite the relevant literature. The developed molecules use a single CDR3 segment to interact with the target, which seems to contradicts with the potential advantage inferred by the quoted statement above on the need for a new scaffold.
2. The developed molecules bind to the site within eIF4E for a known, high-affinity ligand, 4E-BP1, with similar affinity. The two molecules had similar biological effects. Thus, studies with the binder provided little biological insight. Also, because the site is already primed for high-affinity interaction, the results does not provide support the capacity of the developed system to produce high-affinity binders to novel targets.
3. The implicit premise that the presence of a disulfide bond in a scaffold prevents it from intracellular applications is unsupported. The successful application of antibody fragments still containing cysteines, for example the widely used anti-GFP nanobody as well as the so-called intracellular antibodies, demonstrates that the presence of the disulfide bond does not necessarily preclude intracellular applications. As such, the developed scaffold does not offer a clear advantage over existing ones for intracellular applications.
4. The statement that an antibody fragment derived from a human VH domain should have negligible immunogenicity is unsupported, particularly for antibody molecules that have many changes from the germline sequence. For example, Humira, a human antibody, has substantial levels of immunogenicity. These speculations without experimental data should be removed.
5. The manuscript does not describe how the removal of the disulfide bond affects the VH structure. There are no comparisons of the crystal structures with that of the original molecule or related molecules such as nanobodies.
6. The formation of a distinct hydrophobic cluster formed by a long CDR3, revealed in the

structures of the eIF4E binders, is a well-known feature of nanobodies in which such a hydrophobic cluster protects the VH surface that would be used for binding to VL (PMID 23495938). The authors essentially reinvented the design strategy of nanobodies. This point should be clearly documented. The statement, "A unique feature of the evolved eIF4E interacting VH domains is that the randomised CDR3 loop forms a well-defined domain type structure." is unsupported.

Minor points

1. The specificity of the binders need to be better characterized, for example, using affinity purification followed by mass spectrometry-based proteomics.
2. Many figure panels use tiny fonts and they are hard to read. The complete sequences of the binders and raw data for thermal denaturation should be provided in the supplementary information.

Reviewer 1.

General Points:

- 1) **“The title is too broad and should be more specific indicating that an inhibitor of eIF4E was isolated.”**

The title has been amended to “Engineering a Human Derived Autonomous Antibody VH Domain to modulate intracellular pathways and to Interrogate the eIF4F Complex”

- 2) **“The introduction section is very short and very broad and the biological role of eIF4E in general and in malignancy and also the general interplay of eIF4E, eIF4F, eIF4G, eIF4BP1 etc. is not provided. This lack of information makes it very difficult to follow the logic of the in vivo experiments. Details should be given in the introduction section and a scheme/overview would be helpful to be provided at least in the supplemental.”**

More detail on eIF4E is now provided in the introduction. A scheme/overview (Schema S1) has been added to the supplementary information. And referenced in introduction. See below:

“The isolated VH domains were demonstrated to inhibit the eIF4E:4G interaction and were further evolved by affinity maturation to generate a picomolar binder against eIF4E (termed VH-S4) suitable for activity modulation studies in mammalian cells (Schema S1). “

Schema S1: Outline of experimental pipeline from design of an optimized VH framework, generation of phage library, selection against an intracellular target of therapeutic interest (eIF4E), generation of high affinity binder to VH domain evaluation and validation experiments relevant to the selected target.

- 3) **“Another critical point is that the authors claim a new antibody scaffold for in vivo use and insinuate that absence of disulphide bonds is required for folding and intracellular function. This is obviously not (always) the case since numerous nanobodies exist, termed intrabodies that do contain a disulphide bond, are stable in vivo (see e.g. <https://pubmed.ncbi.nlm.nih.gov/33371447/>) and can also be considered for therapeutic**

applications in case that they can be delivered into cells. This should be considered and appropriately cited. “

The reviewer makes several salient points that we have now addressed in the introduction. See below:

Variable domains of the human immunoglobulin heavy chain (VH domains) are ideal candidates for use as mini-proteins. They possess three binding loops of variable length (CDR-H1, CDR-H2 and CDR-H3) that are naturally randomised to generate a wide repertoire of binders for antigen recognition by the immune system. The development of these scaffolds as single domain binding reagents for intracellular studies has been restricted by poor stability, due to the loss of stabilizing interactions with the variable light-chain domain in the intact antibody^{13,14}. However, single chain variable fragments (ScFv) termed “Intrabodies” consisting of a VH and a VL domain connected by a flexible peptide have been used to probe intracellular targets. Unfortunately, ScFvs are deleteriously affected by the reducing conditions of the cell, which prevent the formation of the VH and VL intra-domain disulphide bonds and consequently hinders their proper folding leading to ScFvs that are non-functional, poorly expressed with short half-lives and poor solubilities^{15–18}.

Several approaches have been devised to overcome these liabilities and to increase the discovery rate of intrabodies, such as screening intrabody antigen interactions within cells and the use of predetermined frameworks and consensus sequences with improved solubility and expression properties *in vivo*.¹⁹ Single domain antibody fragments generated from VH and VL domains have also been reported to express within cells and to retain their functionality.²⁰ Consensus sequences derived from these intrabodies were unaffected by substitution of the residues responsible for the VH domain intra-disulphide bond. The dispensability of the intra-disulphide bond is also supported by crystallographic data showing that its absence does not perturb VH and VL domain structures.²¹ An alternative antibody-type modality capable of intracellular expression are Nanobodies (VHH).^{22,23} These are variable domains derived from the heavy-chain only camelid antibodies that are monomeric and soluble that contain 1 to 2 intra-disulphide bonds. However, most Nbs yielded from conventional screening approaches are non-functional within living cells. To overcome these issues, different groups have reported approaches to improve selection of intracellularly functional binders from synthetic or immunized libraries^{22,24,25} and efforts to identify cysteine-free, non-immunoglobulin based miniproteins that are suitable for intracellular expression have also been reported.⁹

Motivated by these observations and by data demonstrating that engineered human VH domain scaffolds can exist monomerically in solution^{13,26}, we endeavoured to overcome the liabilities associated with intracellular expression of ScFvs and other miniproteins by engineering and optimising an autonomous and disulphide-free human VH domain for intracellular expression studies through the use of CoFi²⁷ and Hot-CoFi²⁸ directed evolution techniques.

- 4) **“I also wonder, whether reintroduction of the disulphide bond in VH-S4 results in an equally functional and possibly even more stable variant.”**

This has been addressed, where we see no difference in terms of cellular expression and stability of VH-S4 in mammalian cells. However, we have performed MS experiments that indicate *in vitro* that the intradisulphide bond forms in the C22-C92 containing VH-S4 domain. Additionally, we have also elucidated the crystal complex structure that shows no significant changes in the VH-S4 fold.

Please see the sections entitled **“The presence of C22 and C92 in VH-S4 Is not Detrimental to Function or Intracellular Expression”** and **“The VH-S4 Intra-Disulphide Bond is Retained in a Reducing Environment.”**

Specific Corrections:

- 5) **“I recommend to mutually exchange Figures 1A and B and briefly mention in the text that the Hot-CoFi method is described in Figure 1A. “**

Fig 1A and 1B exchanged and figure resolution improved.

Relevant text changed: **“This was used to initiate 2 rounds of selection using Hot-CoFi²⁸ described in Figure 1A, resulting in the isolation of 53 stabilized VH domain variants (Figure 1B).”**

- 6) “Three residues in VH-36, C22 and 99 C92, as well as the core residue A24, were randomised with all 20 amino acids to generate a library 100 suitable for screening”. Please explain why A24 was selected.

Addressed with the following sentence: “A24 was included to facilitate optimal packing of residues in the hydrophobic core of the protein due to its proximity to C22 and C92.”

- 7) **Li 127: “whilst CDR3 was randomized with sequences of different lengths and biased towards residues serine and tyrosine with approximately 20% frequency each.” Please specify length distribution and how randomization was performed and also how the bias towards Ser and Tyr was achieved.**

This sentence has been replaced with the following:

“A library of VH domains were then displayed on the pIII protein of M13 phage, where the CDR1 and CDR2 of the VH-38i.1 template were randomized conservatively, whilst CDR3 was randomized with sequences of different lengths ranging from 10 to 24 residues and biased towards residues serine and tyrosine with approximately 20-25% and 10-15% frequency, respectively (**Figure S3**), resulting in a library with 2.87×10^{10} unique clones (see methods and materials).”

Figure S3 shows the amino acid frequency analysis for each CDR of the VH-DIF phage display library, as measured by analysing 1,131,370 unique sequences obtained by next-generation sequencing.

The following sentence has been added to the “Phage display library section” in methods and materials:

Fifteen sub-libraries were created by Kunkel mutagenesis with primers containing skewed base compositions, following a similar approach as described by Fuh G. and colleagues [REF

- 8) **How many individual clones were generated upon randomization of CDR1-3?**

Now cited in main text. Relevant changes made. Please see point 7.

- 9) **Phage Display library screening against eIF4E: How many different clones were obtained after how many screening rounds? –**

Additional table (**Table S2**) in supplementary information with clone information. Now also referred to in main text:

“Three rounds of phage display selection against purified human full-length eIF4E, followed by screening of 95 individually-picked phage clones by ELISA led to the identification of 9 unique VH Domains (**Table S2**). VH-1A2 and VH-1C5 were validated as the highest affinity binders with K_{ds} of 115.2 ± 4.4 nM and 154.3 ± 70.3 nM, respectively (**figure 2A, Table 1**).”

Table S2 shown below:

Name	CDR1	CDR2	CDR3	Number of clones	K _d (μM)
1A2	SISSTS	SPSSGSTS	GRMAYSYPSSFSASSFDLYAFD	2	0.044
1A7	SFYITY	YPSNGYTY	GRIYSSAFFYSPTNSYSAFD	3	N.T.
1A8	TFKYTY	FPVLGYTY	GRFSASSYAFD	1	0.244
1A12	FIVSTP	YPFNGSTK	VRSFASYFNPLSVSYSLVAHAVD	1	7.1
1C5	SISSTS	SPSSGSTS	GRVAKDLNSSSPFVVNTYSSFGFD	7	0.078
1C11	YFPHTA	YPYYGTTF	GRYPYTFVGM	1	0.213
1E7	DVYDTA	FPVIGYTS	VRLAYSADFAASEVSSAID	1	3.67
2A3	SIDATY	YPASGITA	GRFFSNPDFAVD	1	0.360
2D4	VVYQTY	YPASGYTY	GRIYASPLYFQNFVFFSNGID	1	15.4

Table S2: List of eIF4E interacting VH domains identified from phage panning experiments against the naïve VH domain library. Selected CDR1,2 and 3 sequences and clone frequency denoted. VH domains were cloned by PCR into pET22B vectors for bacteria protein expression and purification and screened using bilayer interferometry (ForteBio BLItz). Interferometry data was used to determine preliminary dissociation constants (K_d). N.T.: not tested.

Corresponding methodology section has been added to the materials and methods.

10) All Biacore sensorgrams: The concentrations range for measurements or the concentration for curve should be provided.

Concentrations used for SPR measurements are included in the following figure legends:

- Figure 2 legend: "Surface plasmon resonance sensorgrams showing 2-fold titration series of VH-1C5 and VH-1A2 from 2.5 μM to 25 nM against eIF4E immobilised via amine coupling on CM5 chips, respectively (see materials and methods). Binding and kinetic data are shown on the graphs."
- Figure 3 legend: "Surface plasmon resonance sensorgram of VH-1C5^{E104D} being titrated against eIF4E immobilised via amine coupling on a CM5 chip (see materials and methods), using a 2-fold concentration series from 2.5 μM to 2.5 nM. Binding and kinetic data are shown on the graphs."
- Figure 4 legend: "Single injection cycle surface plasmon resonance sensorgrams of VH1C5^{D104A/S108R/F120I}, VH-1C5^{D104A/F120I} and VH-1C5^{D104A/S108R} being titrated across eIF4E immobilised via amine coupling on CM5 chips, using 3-fold concentration series from 0.37 nM to 30 nM (see materials and methods). Binding and kinetic data are shown on the graphs."

- 11) **The function of 4EB1 should be explained upon first mentioning (see above). Likewise: The derived peptide is not cited.**

4E-BP1 is now introduced in the 'introduction'. See below:

"Failure to regulate the eIF4F complex frequently occurs in cancers when the 4E-binding proteins (4E-BP1,2 and 3) are hyperphosphorylated by mTOR^{28,29-31}, and fail to displace eIF4G from eIF4E. Both proteins possess a common primary binding motif (YXXXXLΦ, X = any amino acid and Φ = hydrophobic amino acid) to eIF4E.³²"

Derived peptide is now cited in the following sentence:

"The selected CDR loop sequences of the VH-1A2 and VH-1C5 domains shared little similarity to the shared interacting motif of eIF4E binding proteins such as eIF4G1 and the 4E-BP family (YXXXXLΦ) (figure 2B).³³"

Ref 33: Marcotrigiano, J., Gingras, A. C., Sonenberg, N. & Burley, S. K. Cap-dependent translation initiation in eukaryotes is regulated by a molecular mimic of eIF4G. *Mol. Cell* **3**, 707–716 (1999).

- 12) **Affinity maturation of VH-M4 binder by yeast display is not appropriately described. How many clones were generated? How many were screened? Details should be provided in the supplemental.**

Number of clones generated and their sequences are now clearly described with the following section under the "Affinity Maturation" heading in the supplementary:

*"50 clones were sequenced from each affinity maturation round of selection. The final round of kinetic selection resulted in the identification of only three eIF4E binders: VH-1C5^{D104A/S108R/F120I}, VH-1C5^{D104A/S24G/F120I} and VH-1C5^{D104A/Y97C/F120I}. The VH-1C5^{D104A/S108R/F120I} clone (termed **VH-S4**) dominated the final round with a frequency > 90% and was therefore selected for scale up and SPR analysis. The clone was then amplified using PCR and ligated into the pET22B bacterial protein expression vector for protein purification as outlined in "Bacterial Expression and Purification of VH Domains" section."*

This information is also highlighted in the main text:

*"The final round of kinetic selection resulted in the identification of only three eIF4E binders: VH-1C5^{D104A/S108R/F120I}, VH-1C5^{D104A/S24G/F120I} and VH-1C5^{D104A/Y97C/F120I}. The VH-1C5^{D104A/S108R/F120I} clone (termed **VH-S4**) dominated the final round with a frequency > 90%, which bound eIF4E with a mid-picomolar K_d of 0.057 ± 0.004 nM (figure 4C, Table S3)."*

Table S3 shows the analysis of the sequencing data from each round of selection with associated mutations that was previously shown with a graphic in figure 4.

Referee 2:

Main points.

- 1) How selective is the VH-1C5(D104A/S108R/F120I) evolved clone for eIF4E? For example, if the authors radiolabel cells expressing VH-1C5(D104A/S108R/F120I) with ³⁵S-Met/Cys, and then pull down VH-1C5(D104A/S108R/F120I) they should see eIF4E following SDS-PAGE and autoradiography. Are there other proteins that are pulled down? One might expect to see eIF4A, eIF4G, and/or 4E-BPs depending on the stringency of the buffer used, but other than this – no other proteins should be pulled down. This experiment would provide a sense of what other “targets” might be engaged by VH-1C5(D104A/S108R/F120I). Whatever the answer, it is important to know.

Specificity of VH-S4 has been assessed by performing a FLAG-IP pull-down from cell lysate in comparison to the negative control construct VH-1C5^{scr}. Only eIF4E was detected, whilst no other members of the other eIF4F complex were observed. The following sentence has been added to the main text in the “VH-S4 Disrupts eIF4F Complex Formation and Cap-Dependent Translation In Vitro” section:

“Additionally, the specificity of VH-S4 for eIF4E was demonstrated through anti-FLAG mediated IP of VH-S4 from whole cell lysate, where no other members of the eIF4F complex were detected apart from eIF4E (Figure S9).”

The accompanying figure was inserted in the supplementary as figure S9 with the legend:

“Lysates were prepared from HEK293 cells and were either transfected with mock DNA, VH-S4 or VH-1C5^{scr} and then used to perform anti-FLAG pull down followed by western blot analysis. eIF4G, eIF4A, eIF4E and 4E-BP1 antibodies were used to probe WCL (whole cell lysate), immune-precipitated protein (FLAG-IP) and cell lysate post-immunoprecipitation (post-IP) fractions. Protein levels of VH-S4 and VH-1C5^{scr} were assessed with anti-FLAG”

- 2) Do the authors have a sense of the stoichiometry (at the protein level) of VH-1C5(D104A/S108R/F120I) required to dissociate eIF4E from eIF4G.

From the crystal structure it is apparent that the VH domain interacts with eIF4E via a single binding site with a picomolar affinity as determined by SPR.

- 3) Also, what is the subcellular localization of VH-1C5(D104A/S108R/F120I)? Any VH-1C5(D104A/S108R/F120I) in the nucleus where eIF4E has been implicated in mRNA transport?**

Due to time limitations we have focussed on addressing questions in relation to the effect of the intra-disulphide bond in the VH domain. However, this is a question we plan to pursue in the future.

- 4) Does recombinant VH-1C5(D104A/S108R/F120I) or other variant function in vitro when recombinant protein is added to translation extracts. What is the impact in vitro on cap-dependent versus cap-independent translation in vitro (I know the authors have done the experiments in cells, I am wondering if there is selectivity in vitro).**

Due to time limitations we have focussed on addressing questions in relation to the effect of the intra-disulphide bond in the VH domain. However, the manuscript does attempt to delineate the effects on cap dependent vs cap-independent translation using the bicistronic assay system. Additionally, we also demonstrate the effect of VH domain expression on protein synthesis in cell lines using puromycin incorporation.

- 5) In Fig 5C, please present the FLuc and RLuc values as separate bar graphs. Information is lost when these are presented as ratios. The title of the panel should not be there since there are also results on the IRES in this graph.**

The Fluc and Rluc values are now reported in figure S8 in the supplementary information.

- 6) I think the authors can make the MS a bit more accessible to a broader readership. My suggestion is that the authors revisit the the writing of the paper (at least in some sections). I believe that many labs would be interested in the overall pipeline described by the authors. One area that could use improvement is that they use several specialized technology (and corresponding terms) which are not described in their text.**

Examples are CoFi, Hot-CoFi, DSF, NanoBit [sometimes also spelt NanoBIT in the text], Surefire assay,.... A few introductory sentences as to what these approaches achieve and measure would be helpful here to a reader less knowledgeable with these approaches.

The above points have been addressed with the following sentences:

- a. When Hot-CoFi is initially mentioned in the main text we have included the sentence

“Hot-CoFi is an attractive system for the selection of thermally stable VH variants as high yield bacterial expression has been shown to correlate well with ScFvs expression in the cytoplasm of mammalian cells.¹⁹”

- b. When CoFi is introduced we have added the explanatory sentence “*CoFi screening as opposed to Hot-Cofi was performed at room temperatures to identify soluble and colloiddally stable variants.*²⁷”

- c. DSF (differential scanning fluorimetry) is a much more common technique than the CoFi methods and is only used for measuring protein stability. We feel it needs no further explanation. For this we have therefore only added:

“ see material and methods” and an extra reference in the following sentence: “These were purified, and their thermal denaturation point (T_m) determined by differential scanning fluorimetry (DSF³⁵, see methods and material), with VH-36i.1 demonstrating the highest thermal stability (T_m of 53.2°C). “

- d. The NanoBiT assay is now introduced with the sentence:
“The NanoBit assay consists of a split luciferase fused either to eIF4E or eIF4G⁶⁰⁶⁻⁶⁴⁶, whose interaction restores luciferase function.^{41,60} Exogenous expression of antagonists of the eIF4E:4G interaction result in a decrease in total luminescence.” in the figure 5 legend.
- e. When the Surefire assay is referenced in the figure 5 legend, the following sentence has been added:
- a. *“This assay is a no wash sandwich bead-based immunoassay that enables rapid detection and precise quantification of eIF4E phosphorylation levels.”*
- f. *“I ask the that nucleotide sequences of the different engineered mini-proteins (especially VH-S4) be included in an Appendix. Without these it becomes difficult for labs to reproduce and build on this nice piece of work. The amino acid sequence is a start, but codon usage or bias cannot be assessed from solely amino acid sequence.”*

VH sequences have been included in the supplementary section in table S5.

Referee 3

Main points:

1. This system joins a sizable group of already established molecular scaffolds that can be used for intracellular application, including nanobody, monobody and DARPin to name just a few most advanced systems (PMID 20010839, 33371447, 32145686 and 32591521 and references therein). As such the novelty of the system itself is limited. The statement, “Many currently used miniproteins are limited as binding modules due to randomization of highly rigid structural motifs with limited sites of variation.” (page 2) seems to contradict with the large body of the literature. The manuscript does not cite the relevant literature. The developed molecules use a single CDR3 segment to interact with the target, which seems to contradict with the potential advantage inferred by the quoted statement above on the need for a new scaffold.

The following section has been removed:

“Additionally, they should have negligible issues with immunogenicity issues in human therapy. However, the development of these scaffolds as single domain binding modalities has been restricted by poor stability, due to the loss of stabilizing interactions with the light-chain in the intact antibody^{6,7}. Several groups have identified monomerically stable VH domains either by serendipitous discovery or by using phage-based evolution methods with the VH domain of trastuzumab (4D5) as a template.^{6,8} Despite the progress in the generation of an autonomous, stable VH domain, there has been no success in generating a stable disulphide-free VH domain (DiF-VH), which can be used for intracellular applications.”

It has been replaced with a section describing why it was decided to develop a disulphide free VH domain more clearly by outlining the following points below:

- Current status of single chain antibodies (ScFvs) for intracellular expression “intrabodies”
- The potential liabilities of cysteine.
- Removal of cysteine’s have little effect on structure and expression of “Intrabodies”. Many of which have been discovered serendipitously or using in vivo selections.
- The potential of these to be expressed as single domain
- Current strategies to alleviate intracellular expression issues.
 - Consensus sequence
 - In vivo selections
 - Use of NanoBodies
- The existence of VH domains that are monomerically stable in vitro
- A sentence describing other non-antibody based modalities
- The decision to develop disulphide free VH domain to avoid liabilities associated with cysteine and intracellular expression, and additionally to use this scaffold as a basis for a high diversity phage library. A strategy that offers a discrete advantage over lower diversity selection performed in vivo. This should lead to the higher discovery rates of human VH domains suitable for intracellular experiments.

See below for added text corresponding to above points:

“Variable domains of the human immunoglobulin heavy chain (VH domains) are ideal candidates for use as mini-proteins. They possess three binding loops of variable length (CDR-H1, CDR-H2 and CDR-H3) that are naturally randomised to generate a wide repertoire of binders for antigen recognition by the immune system. The development of these scaffolds as single domain binding reagents for intracellular studies has been restricted by poor stability, due to the loss of stabilizing interactions with the variable light-chain domain in the intact antibody^{13,14}. However, single chain variable fragments (ScFv)

termed “Intrabodies” consisting of a VH and a VL domain connected by a flexible peptide have been used to probe intracellular targets. Unfortunately, ScFvs are deleteriously affected by the reducing conditions of the cell, which prevent the formation of the VH and VL intra-domain disulphide bonds and consequently hinders their proper folding leading to ScFvs that are non-functional, poorly expressed with short half-lives and poor solubilities^{15–18}.

Several approaches have been devised to overcome these liabilities and to increase the discovery rate of intrabodies, such as screening intrabody antigen interactions within cells and the use of predetermined frameworks and consensus sequences with improved solubility and expression properties *in vivo*.¹⁹ Single domain antibody fragments generated from VH and VL domains have also been reported to express within cells and to retain their functionality.²⁰ Consensus sequences derived from these intrabodies were unaffected by substitution of the residues responsible for the VH domain intra-disulphide bond. The dispensability of the intra-disulphide bond is also supported by crystallographic data showing that its absence does not perturb VH and VL domain structures.²¹ An alternative antibody-type modality capable of intracellular expression are Nanobodies (VHH).^{22,23} These are variable domains derived from the heavy-chain only camelid antibodies that are monomeric and soluble that contain 1 to 2 intra-disulphide bonds. However, most Nbs yielded from conventional screening approaches are non-functional within living cells. To overcome these issues, different groups have reported approaches to improve selection of intracellularly functional binders from synthetic or immunized libraries^{22,24,25} and efforts to identify cysteine-free, non-immunoglobulin based miniproteins that are suitable for intracellular expression have also been reported.⁹

Motivated by these observations and by data demonstrating that engineered human VH domain scaffolds can exist monomerically in solution^{13,26}, we endeavoured to overcome the liabilities associated with intracellular expression of ScFvs and other miniproteins by engineering and optimising an autonomous and disulphide-free human VH domain for intracellular expression studies through the use of CoFi²⁷ and Hot-CoFi²⁸ directed evolution techniques. The optimised VH framework sequence was then used to generate a phage display library with randomised CDR1, 2 and 3 loops, that was used to identify several VH domain binders against eIF4E. The isolated VH domains were demonstrated to inhibit the eIF4E:4G interaction and were further evolved by affinity maturation to generate a picomolar binder against eIF4E (termed VH-S4) suitable for activity modulation studies in mammalian cells (**Schema S1**). Failure to regulate the eIF4F complex frequently occurs in cancers when the 4E-binding proteins (4E-BP1,2 and 3) are hyperphosphorylated by mTOR^{29,30–32}, and fail to displace eIF4G from eIF4E. Both proteins possess a common primary binding motif (YXXXXLΦ, X = any amino acid and Φ = hydrophobic amino acid) to eIF4E.³³ The identified high affinity VH domain shares an overlapping interaction site with peptides derived from this binding motif but interacts through a unique binding mode. The interface between eIF4G and eIF4E is a potential site of therapeutic development and has been targeted with several small molecule and peptidic strategies.^{6,34} However, many of these molecules are poorly active and have not been clinically approved and the novel interaction pose formed by VH-S4 represents an alternative and unexplored avenue for rational therapeutic development strategies.

The autonomous VH domain technology presented here offers a rapid and efficient pipeline to discover new binding poses for therapeutic lead development, as well as the identification of modalities that can be used to therapeutically model and validate these sites for drug development in biological systems. This methodology also enables the advantageous linkage of high diversity *in vitro* libraries (10⁸-10¹⁰) with an intracellularly expressible mini-protein scaffold, in contrast to *in vivo* selection techniques that use smaller libraries (10⁵-10⁶), to increase the discovery rate of VH domains suitable for intracellular applications.”

One point we would like to highlight is that the phage libraries are randomised at all 3 CDR position. And it is only the selection process against eIF4E that results in the isolation of 2 validated VH domains that interact with eIF4E primarily through the CDR3 region.

2. The developed molecules bind to the site within eIF4E for a known, high-affinity ligand, 4E-BP1, with similar affinity. The two molecules had similar biological effects. Thus, studies with the binder provided little biological insight. Also, because the site is already primed for high-affinity interaction, the results does not provide support the capacity of the developed system to produce high-affinity binders to novel targets.

I disagree with these comments:

- 1) The site overlaps with the eIF4G and 4E:BP1 binding sites and is therefore non-identical.
- 2) It is a dramatically different binding pose that offers a very different pharmacophore for drug design and discovery. For example, a dramatic decrease in positive charge is required for the VH-S4 interaction and the eIFG1 motif is mimicked by a distinctly different structural element.

- 3) Therefore, validation of this site as an alternative site of inhibition, especially as it is a clearly different to other approaches that use molecules to mimic the helical conformation of the 4G and 4E-BP1 peptides (such as stapled peptides) is particularly useful.
- 4) The fact that these molecules have similar biological function is highly re-assuring as this validates the pharmacophore for therapeutic development. It would be high undesirable if it did not.
- 5) Additionally, by having a high diversity library that has isolated a previously unseen new interaction mode at this well studied site, I would argue is more than enough and sufficient to validate this library and approach.

3. The implicit premise that the presence of a disulfide bond in a scaffold prevents it from intracellular applications is unsupported. The successful application of antibody fragments still containing cysteines, for example the widely used anti-GFP nanobody as well as the so-called intracellular antibodies, demonstrates that the presence of the disulfide bond does not necessarily preclude intracellular applications. As such, the developed scaffold does not offer a clear advantage over existing ones for intracellular applications. These speculations without experimental data should be removed.

This is a well-made point and there are several examples of this. However, there is also a reasonable body of work describing otherwise. Additionally, this is also true for NanoBodies, where most Nbs identified with conventional screening approaches have behaved poorly within living cells, and as with ScFvs there are several exceptions to this rule such as the anti-GFP nanobody mentioned by the reviewers amongst others. Also, as with ScFvs, there are several similar approaches for potentially addressing this issue. We have therefore re-written the introduction to emphasise that the VH domain and the associated phage library have been developed to increase the rate of discovery of useful VH domains that are amenable to intracellular expression, bearing the above points in mind. **See text in point 2.**

Furthermore, we have performed extra experiments to delineate the role of the two cysteines in the VH domain, primarily by removing them or re-introducing them to the variants identified at various stages of the directed evolution process, including the final selected VH-S4 domain. Interestingly, the overall increase in thermal stability of the monomeric VH domain correlates with improved cellular expression and is not dependent on the presence of the C22-C92 intra-disulphide bond. Additionally, the presence or absence of these residues do not impair VH-S4 functionality. Interestingly, the expression of the VH-37i variant that express poorly in absence of the C22 and C92 residues, has its expression rescued upon their re-insertion. These results suggested to us that the intra-disulphide bond is present in the reducing environment of the cell. We, therefore performed comparative mass spectrometry experiments that identified the presence of an intra-disulphide bond between C22 and C92. Additionally, we also elucidated the crystal structure of VH-S4 with a disulphide bond present bound to eIF4E to demonstrate it has a negligible on the overall fold of the VH domain.

These results are described in the 2 sections below (added at the end of the results section):

The presence of C22 and C92 in VH-S4 Is not Detrimental to Function or Intracellular Expression

With the VH-S4 domain demonstrating sufficient mammalian cellular expression to perturb eIF4F function, we assessed the effects of reintroducing C22 and C92 into VH-S4 upon its performance, whereupon negligible differences in cellular expression, proteosomal stability and in the amounts of eIF4E immune-precipitated were observed between the two variants (**Figure 7A and 7B**). This prompted further examination of the differences in mammalian expression levels of the various VH domain mutants evolved at different stages of the directed evolution process (**Figure 7C**) with or without C22 and C92. From

these results, it was apparent that the 4D5 template sequence with or without C22 and C92 was suboptimal for cellular expression and that improvements in the thermal stability of the VH variants generally correlated with increased soluble mammalian cellular expression (**Figure 7C** and **Figure S2**). Incorporation of C22 and C92 predominately increased the thermal stability of the purified VH domains by approx. 10°C indicative of the formation of intra-domain disulphide bond formation *in vitro*. However, the formation of this bond in the cytoplasm is difficult to establish but it is apparent that the other mutations in VH-36 and later variants contributed to the soluble and stable expression of the VH domain constructs intracellularly in the absence of C22 and C92. The replacement of the previous CDR3 loop in VH-37i resulted in loss of cellular expression and thermal stability that was either compensated for by the re-introduction of C22 and C92, suggesting that these residues are forming stabilising interactions with each other inside the cell, or by the incorporation of S93V and A78V in VH-37i.1 and VH-37i.2, respectively, that both facilitate improved scaffold stability.

These observations then lead us to demonstrate that the C22-C92 residues form a disulphide bond in a reducing environment. See section below.

The VH-S4 Intra-Disulphide Bond is Retained in a Reducing Environment.

The rescue of intracellular expression of VH-37i by the re-insertion of C22 and C92 raises the possibility of intra-disulphide bond formation occurring inside the cell, especially with both residues residing internally near each other in the VH domain and occluded from the reducing cellular environment. To establish the presence a disulphide, recombinant VH-S4(S22C-T92C) was analysed by liquid chromatography/mass spectrometry after treatment with defined redox conditions (**Figure 7D**). Treatment of VH-S4(S22C-T92C) with the reducing agent DTT and the chemical denaturant guanidinium chloride lead to a molecular weight shift of approximately 2 Da, consistent with the loss of an intra disulphide bond. As expected from literature precedent,^{53,54} disulphide reduction was accompanied by a shift in the spectral envelope toward higher charge states, providing a second line of evidence that a disulphide had been present before reducing/denaturing treatment. No significant shift in mass or spectral envelope was measured after treatment with DTT in non-denaturing conditions, suggesting the disulphide would be stable intracellularly. Together this data suggests that recombinant VH-S4(S22C-T92C) contains a disulphide bond that is preserved in the reducing environment of the cell. VH-S4(S22C-T92C) was also crystallised in complex with eIF4E, where the presence of the intra-disulphide bonds caused no conformational changes in the global fold of the VH domain compared to the disulphide free domain (**Figure S11**). Interestingly, the S108R sidechain is found forming a direct electrostatic interaction with E132 and displacing the structured waters observed in the VH-S4 structure (**Figure 4D** and **Figure S11**).

Additionally, we made the following changes to the 1st paragraph of the conclusion outlying the potential advantages of C22-C92 reinsertion into the scaffold after phage selection.

We have developed a repertoire of VH domain-based mini-proteins that sample a wide range of K_{d} s (picomolar to micromolar), which disrupt the eIF4F complex through a novel binding pose and that are amenable to cellular expression. We have also established that the residues (C22-C92) required for intra-disulphide bond formation are not detrimental to the expression of the majority of VH variants with improved thermal stabilities (**Figure 7C**) and that these residues form a disulphide bond when re-substituted into the VH-S4 domain in the presence of a reducing environment. The re-introduction of the intra-disulphide bond also has no effect on the VH domain structure or its interaction with eIF4E (**Figure 7** and **Figure S11**). Further, the re-substitution of C22 and C92 into VH-37i restored its cellular expression in the absence of other VH variant mutations (VH-37i.1 and VH-37i.2) that thermally stabilized the VH domain (**Figure 7C**). These results suggest that if poorly expressing VH domains transiently form correctly folded structures upon protein synthesis, then occlusion of C22 and C92 from the reducing environment will enable disulphide formation and allow improved cellular expression. This suggests that in phage selections utilizing the disulphide-free VH scaffold, which fail to identify a clone amenable to cellular expression, re-substitution of C22 and C92 is a potential rescue strategy.

3. The statement that an antibody fragment derived from a human VH domain should have negligible immunogenicity is unsupported, particularly for antibody molecules that have many changes from the germline sequence. For example, Humira, a human antibody, has substantial levels of immunogenicity. These speculations without experimental data should be removed.

Removed. This is indeed speculative and hypothetical.

4. The manuscript does not describe how the removal of the disulfide bond affects the VH structure. There are no comparisons of the crystal structures with that of the original molecule or related molecules such as nanobodies

The following statements have been added at the relevant points in the text with response to the underlined section with associated changes in the referenced figure to highlight them:

- a. *“VH-36 variant contained the following mutations, H33D, S93G and W103R. Both H33D and W103R are located at the former VH and VL domain interface and mainly improve thermal stability of the VH variant by decreasing its hydrophobicity (Figure S1).”*
- b. *“VH36i.1 possessed the mutations C22S, A24C and C92T (Figure 1C and 1D, Table S1). Hydrogen bonding between C22S and C92T most likely occurs to replace the lost disulphide bond.”*
- c. *“Both clones contained a single mutation with VH-37i.1 bearing a A78V mutation in the core of the VH domain, whilst VH-37i.2 contained the mutation G93V located near the CDR-H3 loop (Figure 1D, Table S1). The introduction of A78V most likely leads to improved packing interactions in the hydrophobic core of the protein around C22S and C92T, whilst the S93V substitution likely increases the rigidity of the β -strand leading into the CDR3 region and thus stabilising the VH framework also.”*
- d. *“The VH-38i variant VH38i.1, containing mutation C24I was the most stable with improved T_m of 62.8 °C (Figure 1C). Replacement of C24 with these residues lead to further optimised hydrophobic packing of residues around C22S and C92T (Figure 1D).”*

5. There are no comparisons of the crystal structures with that of the original molecule or related molecules such as nanobodies.

See section 6) below. In response to the above comment as part of the discussion on NanoBodies we have now included a supplementary figure comparing the structure of VH-S4 with several NanoBody structures. In this figure we highlight the similarities in the CDR3 loop interactions and presentation.

6. The formation of a distinct hydrophobic cluster formed by a long CDR3, revealed in the structures of the eIF4E binders, is a well-known feature of nanobodies in which such a hydrophobic cluster protects the VH surface that would be used for binding to VL (PMID 23495938). The authors essentially reinvented the design strategy of nanobodies. This point should be clearly documented.

The clarity of this section has been improved further with the following edits:

“These residues form a distinct hydrophobic cluster that interact with several hydrophobic residues located on the β -sheet face of the VH domain (V39, L47 and W49) on the former VL interface (figure 3E). This is highly similar to CDR-H3 loop interactions that occur in a variety of Nanobodies, where an extended CDR3 loop folds back on to the protein to interact with and shield hydrophobic residues from the solvent (Figure S5).^{22,39,40”}

7. The statement, “A unique feature of the evolved eIF4E interacting VH domains is that the randomised CDR3 loop forms a well-defined domain type structure.” is unsupported.

This sentence has been removed. And is now only referred to in **point 6)**

Minor comments from all 3 reviewers have been addressed.

Reviewers' Comments:

Reviewer #1:

Remarks to the Author:

The revised manuscript improved significantly and gained clarity. Also the experiment suggested by the reviewer was performed. The paper is now clearly structured and informative and I can recommend it for publication.

Reviewer #2:

Remarks to the Author:

I am satisfied with the authors' responses to my concerns.

Reviewer #3:

Remarks to the Author:

This revised manuscript has constructively addressed most of points raised for the previous version. The added data support the revised conclusions, and unsupported statements have been properly modified or removed. This work offers diverse information of interest.

The following, relatively minor points should be addressed.

L.75-76. A reference needs to be added for "However, most Nbs yielded from conventional screening approaches are non-functional within living cells."

L.121. "TCAGGS" needs to be defined.

L.174. "The crystal complex of m7GTP and VH-1C5 (Figure S4)" differs what is shown in Figure S4.

L.187. "that precedes the loop re-joining the VH framework". Unclear.

L.339 and throughout. "intra-disulfide bond". Do the authors mean "intra-domain disulfide bond"?

Figure 7C. The MS data need to be properly labeled.

Supplementary Data. p. 37. The section, "VH-S4ss condition + details" seems to be missing text.

Dear Editor

Here are the changes we have made to the manuscript in response to the points raised by the reviewers:

Reviewer 3.

1) L.75-76. A reference needs to be added for “However, most Nbs yielded from conventional screening approaches are non-functional within living cells.”

The following references have been inserted.

“Nanobodies Right in the Middle: Intrabodies as Toolbox to Visualize and Modulate Antigens in the Living Cell”, Teresa R. Wagner and Ulrich Rothbauer. *Biomolecules*, 2020 Dec; 10(12): 1701. Published online 2020 Dec 21. doi: [10.3390/biom10121701](https://doi.org/10.3390/biom10121701)

And

“A general approach for stabilizing nanobodies for intracellular expression”. J Dingus, CY Tang and C Cepko. *bioRxiv*, 2021.04.06.438746; doi: <https://doi.org/10.1101/2021.04.06.438746>

2) L.121. “TCAGGS” needs to be defined.

The term TCAGGS has now been defined see below in bold:

“Among the identified variants, VH-33 and VH-36 were the most thermally stable, with T_{CAGGS} (**midpoint of thermal cellular aggregation curves**, see materials and methods) of 72.9°C and 73.4 °C, respectively, an increase of over 20 °C in comparison to the 4D5 clone (**Figure 1C, Table S1**). “

3) L.174. “The crystal complex of m7GTP and VH-1C5 (Figure S4)” differs what is shown in Figure S4.

This was a typo and has been corrected. See below”

“The crystal complex of eIF4E and VH-1C5 (Figure S5).....”

4) L.187. “that precedes the loop re-joining the VH framework”. Unclear.

This has been changed to the following for greater clarity. The change has been highlighted in bold

F220, positioned on the short helical turn motif (residues 119-121) **that precedes the loop re-joining the VH domain**. This should make it clearer the CDR3 loop is re-joining the main body of the VH protein

5) L.339 and throughout. “intra-disulfide bond”. Do the authors mean “intra-domain disulfide bond”?

Intra-disulphide bond now has now been replaced with intra-domain disulphide bond throughout the manuscript.

6) Figure 7C. The MS data need to be properly labeled.

The figure has now been appropriately labelled. See below:

7) Supplementary Data. p. 37. The section, “VH-S4ss condition + details” seems to be missing text.

Crystallization conditions for VH-S4ss in complex with eIF4E have now been added to the “Protein Crystallization” methods section. See below:

“Crystals containing the m⁷GTP:eIF4E:VH-S4ss complex were isolated in 0.1M Sodium HEPES pH7.5, 25% PEG 6000. For X-ray data collection at 100 K, crystals for all sets of crystallization conditions were transferred to an equivalent mother liquor solution containing 25% (v/v) glycerol and then flash frozen in liquid nitrogen.”

Regards

Christopher Brown